# Exploring the role of stromal osmoregulation in cancer and disease using executable modelling

David Shorthouse [1], Angela Riedel[1], Emma Kerr[1], Luisa Pedro[1], Dóra Bihary[1], Shamith Samarajiwa [1], Carla P. Martins[1], Jacqueline Shields[1] & Benjamin A. Hall[1]

Osmotic regulation is a vital homoeostatic process in all cells and tissues. Cells initially respond to osmotic stresses by activating transmembrane transport proteins to move osmotically active ions. Disruption of ion and water transport is frequently observed in cellular transformations such as cancer. We report that genes involved in membrane transport are significantly deregulated in many cancers, and that their expression can distinguish cancer cells from normal cells with a high degree of accuracy. We present an executable model of osmotic regulation and membrane transport in mammalian cells, providing a mechanistic explanation for phenotype change in varied disease states, and accurately predicting behaviour from single cell expression data. We also predict key proteins involved in cellular transformation, *SLC4A3* (AE3), and *SLC9A1* (NHE1). Furthermore, we predict and verify a synergistic drug combination in vitro, of sodium and chloride channel inhibitors, which target the osmoregulatory network to reduce cancer-associated phenotypes in fibroblasts.

[1] MRC Cancer Unit, University of Cambridge, Hutchison/MRC Research Centre, Box 197, Cambridge Biomedical Campus, Cambridge CB2 0XZ, UK. Correspondence and requests for materials should be addressed to J.S. (email: js970@mrc-cu.cam.ac.uk) or to B.A.H. (email: bh418@mrc-cu.cam.ac.uk)

Osmotic regulation is necessary for the maintenance of cell integrity under a wide range of conditions. Through the conservation of a robust equilibrium cells can avoid the bursting and destruction of cell membranes caused by extreme, rapid shrinking and swelling[1]. Cells can respond to hypertonically induced shrinking or hypotonically induced swelling by altering the balance of channels and transporters in the extracellular and organellular membranes to manipulate water and solute flow, whilst maintaining cell size[2]. Changes in ion and osmolyte flow result in osmotic pressure, which leads to rapid entry or exit of water through pores such as aquaporins[3].

To combat osmotically induced swelling or shrinking, cells initially activate or alter the expression of pumps, channels, or transport proteins associated with ion flux[4] before stabilising ion concentrations with organic osmolyte transport. This enables a cell to maintain size and reduce osmotic pressure by extruding or importing ions, whilst preserving electrochemical gradients[2,4–6]. Osmotic regulation is highly conserved in mammals and involves a relatively small number of proteins that respond directly to osmotic pressure[7]. Whilst the signalling mechanisms of osmoregulation are highly complex, key ions and proteins involved in the primary response are recognised to be sodium, potassium, chloride, and to a lesser extent, calcium[2,8–11]. Disruption to such tight regulation due to aberrant transporter expression is associated with pathologies such as cancer[12–15] and generally results in changes in cellular morphology, particularly because the principal channels involved in osmotic regulation influence cellular behaviour in ways separate from purely maintaining cell size.

Computational network modelling is a technique for studying the interconnected networks of genes and proteins involved in cellular decision making that is distinct from traditional mathematical modelling[16,17]. In computational (also called executable) modelling, nodes representing genes, proteins, chemical components, or abstract concepts (such as the pressure felt by a cell) have a finite set of discrete values (for example, integers from 0 to 5) representing their activity, concentration, or expression. A key advantage of this methodology is the ability to model in the absence of precise kinetic data, and the ability to exclude missing links, where intermediates are unknown. Additionally, executable modelling allows the use of model checking techniques[18], initially developed for software engineering, that allows analysis of the complete behaviour of the system (e.g., State X can never occur, condition Y always leads to state Z), even in systems with millions of states.

Whilst ion channels have previously been studied from a network modelling perspective, these have generally been limited to highly specialised models of single channel activity[19], or models of current changes in specific tissue subtypes[20–22]. Moreover, extensive previous work on modelling osmoregulation has been performed in yeast cells, but this has focused on the protein signalling cascades behind glycerol synthesis[23–25], rather than the primary ionic response.

Here, we show, firstly, that ion channels and osmoregulatory transport proteins are a marker of cancer phenotype though a machine learning classification approach. Using publicly available data on the expression of membrane protein transporters and channels in cancer, we show that membrane transport proteins are a good descriptor of whether a cell is from a cancer associated sample or not, and when we extract weightings describing which proteins contribute significantly to this classification, top contributors to this classification are transporters involved in osmotic regulation. We then present an executable model of the integrated network of ion channels involved in osmotic regulation and other interlinked channel-dependent processes, including calcium signalling, pH maintenance, and metabolic transport. The network

model predicts cellular behaviour from single-cell expression data, and is a scaffold for developing a greater understanding of drug effects on emergent cellular behaviour, predicting drug/protein targets, predicting measurable changes in detectable substances (e.g., metabolites), and generating limited mechanistic understandings of cellular behaviour. The resultant network predicts cellular phenotype changes from single cell gene expression data, and specific predictions on the effects of channel knock-outs are validated in vitro. Additionally, synergistic drug combinations that target *SLC9A1* (NHE1), and *SLC4A3* (AE3), are discovered by the model and validated in vitro. We applied our model to single-cell gene expression data from two systems of interest: first lymph node stromal cells under two highly distinct physiological conditions a) exposure to tumour-derived fluid, and b) exposure to the bacterial infection stimulus LPS, and second to mouse embryonic fibroblasts (MEFs) exhibiting heterozygous or homozygous mutations in the *Kras* gene.

## Results

**Membrane transport is altered in cancer**. We identified the membrane transport regulatory network for study due to its ubiquity, and the proposed impact ion transport has on many of the cell phenotypes involved in cancer and disease such as cellular proliferation and migration[12,13,26,27]. Genes were selected for study from the genenames database (www.genenames.org), selecting for genes involved in membrane transport. A resultant gene list of 380 genes was curated (Supplementary Data 1), and all the publicly available RNA seq read counts were downloaded and processed from TCGA (The Cancer Genome Atlas) as described in the methods. The resultant dataset contains 11,574 gene expression profiles, from 37 separate projects, divided between 15 subtypes of cancer. We initially clustered the dataset with use of t-Distributed Stochastic Neighbour Embedding (t-SNE) (Fig. 1a). Individual samples are coloured by germ layer origin (mesoderm, ectoderm, endoderm, mesoderm/ectoderm, or unknown). For visualisation purposes, we applied k-means clustering to the t-SNE distribution, and calculated the Voronoi surface based on the k-means centres. The resultant plot contains significant clusters that are broadly of the same germ layer origin. Additionally, most Voronoi cells contain mainly samples of a similar or related cancer origin (for full origins of each cell within the plot see (Supplementary Fig. 1A)). Highlighting samples labelled as non-cancerous within the dataset results in clustered regions of non-cancer samples together.

Due to the ability of ion channel expression data to discriminate between both organ/tissue types, and cancer/non-cancer cells, we chose to further explore the data with the use of binary classification, specifically gradient boosted decision trees. We selected the algorithm XGBoost, due its usability, ability to handle imbalanced datasets, high performance in Kaggle competitions, and interpretability. We found that we were able to classify cells labelled as cancer/non-cancer with very high accuracy—98.92% were correctly classified, with a Matthews Correlation Coefficient (MCC) of 0.91 (Table 1). A comparative sized set of random genes that exclude channels (Supplementary Data 2) generated an MCC score of only 0.81 due to a significantly higher false negative rate. We also applied binary classification techniques to subsets of the data based on cellular germ layer origin, and found similar classification accuracy on subsets of the data (Table 1).

Additionally, exploring the feature weights for the model allows easily human-interpretable rankings of genes deemed most important for the classification. Feature weights reveal genes that were used the most to discriminate between cancer and non-cancer samples (Fig. 1b). Feature weights are shown for the cells

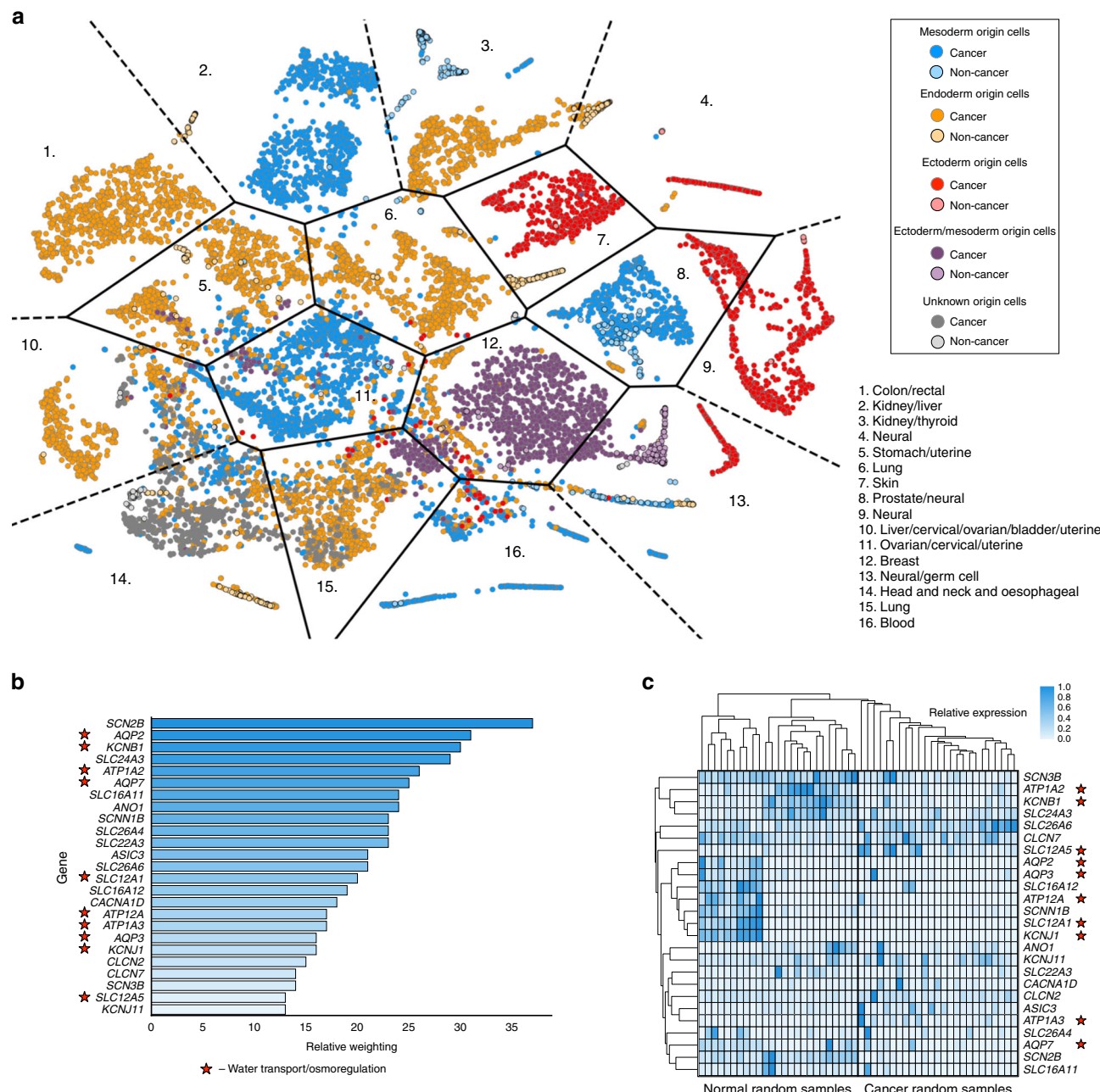

**Fig. 1** Analysis of TCGA gene expression data. **a** Using the t-SNE dimension reduction technique applied to ion channels/membrane protein gene expression alone we show that samples cluster into subgroups classified by their cellular origin (blue—mesoderm, orange—endoderm, red—ectoderm, purple—ectoderm/mesoderm, grey—unknown), and whether they are from a sample determined to be cancer or non-cancer (dark vs. light colours). We overlay the plot with a Voronoi surface calculated from k-means centres. Within each Voronoi tile is a number, relating to the legend, which denotes the majority tissue/organ of origin of each sample in that tile. Applying a binary classification machine learning method to classify cancer vs. non-cancer sample types results in high classification accuracy. Feature weights are then extracted from the model. **b** Shown are the top 25 weighted features for subsamples of mesoderm origin, in particular, there are a large number of genes encoding proteins involved in movement of water—aquaporins, or involved in the osmotic response such as *SLC12A1*, encoding the sodium-potassium-chloride symporter NKCC2 (starred). **c** Taking 25 randomly selected samples of non-cancer and cancer subtypes we plot the expression of the 25 top weighted features described previously, and show that even within a small random subset of the data, differences can be discerned in the expression of these genes between non-cancer and cancer cells

of mesoderm layer origin, for other origins see Supplementary Fig. 1B. Highly featured proteins include numerous proteins involved in osmoregulation or water transport (indicated by a star—10 out of the top 25 weighted features for mesoderm germ origin samples), and many known to be involved in cancers, such as *SLC12A5*[28,29], and numerous aquaporins[30]. Additionally, heatmaps generated from random subsets samples of the datasets

confirm that there are significant differences between non-cancer and cancer expression of membrane transport genes (Fig. 1c, and Supplementary Fig. 1c). Due to the demonstrated importance of membrane transport, and in particular osmoregulation in human tissue data, we chose to study the network of osmoregulatory and membrane transport proteins in general with the use of an executable model.

**Table 1 Results from binary classification analysis of 11,574 samples from the TCGA dataset**

| Data subset | Accuracy | Accuracy (%) | False positives | False negatives | MCC score |
|---|---|---|---|---|---|
| All samples | 2290 of 2315 | 98.92 | 2 of 2159 | 23 of 156 | 0.91 |
| EC origin cells | 558 of 564 | 98.94 | 0 of 535 | 6 of 29 | 0.89 |
| EN origin cells | 849 of 857 | 99.07 | 1 of 790 | 7 of 67 | 0.93 |
| M origin cells | 1015 of 1030 | 98.54 | 3 of 944 | 12 of 86 | 0.9 |
| Random non-channel subset | 2221 of 2315 | 95.94 | 13 of 2006 | 81 of 309 | 0.81 |

Data are classified using only expression data of ion channels, and shows a high accuracy when compared to a set of random non-channel genes from the same dataset (Matthews Correlation Coefficient of 0.91 vs. 0.81). Included are the false positive numbers and false negative numbers

**Table 2 Formal specification for base osmotic regulation model**

| Environmental responses | Expected change | Reasoning | Model expected results |
|---|---|---|---|
| Cell in non-stimulating environment | Network stability | Cell maintains constant size under homestatic conditions (McManus et al.[72], Ho[4]) | Network is stable with external ionic concentration of 2 |
| Cell response to low osmotic environment | Increase in activation state of efflux channels, network stability | McManus et al.[72], Ho[4] | Low extracellular ions activates efflux channels. Network stabliity |
| Cell response to high osmotic environment | Increase in activation state of influx channels, network stability | McManus et al.[72], Ho[4] | High extracellular ions activates influx channels. Network stability |
| Cell response to varied osmotic environments | Cell remains the same size despite changes in osmotic environment, any change in cell size is transient | Cell maintains constant size after being subjected to reasonable osmotic fluctuations (McManus et al.[72], Ho[4]). | Network is stable with external ionic concentration of 0 or 4 |

Shown is the expected cellular behaviour under numerous conditions, the reasoning, and the expected results from the model

**A model of cellular osmotic response**. We identified the osmotic regulatory network for further study based on the high prevalence of osmotically involved transporters in the analysis of publicly available gene expression data. Based on the literature we generated a formal specification (Table 2) that describes the expected behaviour of a model based on the reported primary mammalian cell osmotic response[1,5,6]. A qualitative network model (QN) was generated by manual curation of the literature and represents the canonical set of eight genes and proteins involved in the mammalian primary osmotic regulation. Specifically, we focused on modelling channels as activators or inhibitors of the ion or molecule that they transport.

The resultant model contains 20 nodes representing proteins or cellular components (schematically shown in Fig. 2a, shown in the BioModelAnalyzer (BMA) interface in Supplementary Fig. 2). We include abstracted variables for Osmotic Pressure, representing the pressure exerted on the cell due to the imbalance in external and internal ions, a node for External Ions with a range of 0–2 (0—low, 1—normal, and 2—high), signifying a change or alteration in the osmolarity of the solute surrounding a cell, and a node for cell size that responds to the osmotic pressure within the cell, representing the flow of water entering or exiting the cell.

In this model, functionally assigned nodes were considered to have a range of 0–4, representing minimal activity/concentration (0), lowered activity/concentration (1), normal activity/concentration (2), elevated activity/concentration (3), and maximal activity/concentration (4). Biologically, proteins involved in ionic transport are activated through various cascade mechanisms, and in some cases are manufactured in response to osmotic stress, meaning that they cannot respond instantly to changes in osmotic pressure[31,32], and there is a small lag time before differences in osmotic pressure are rectified. Timing differences in the system were explicitly considered (See Supplementary Note 1).

Homoeostasis was confirmed with the use of stability analysis. Stability indicates that there is a single, global, fixed point the

model returns to regardless of perturbations or initial state. The model re-establishes a stable state at a normal cell size after an initial period of swelling or shrinking when external osmotically active ion concentrations are perturbed. When the cell is exposed to high external ion concentrations, it initially shrinks as water is pulled out of the cell, before returning to normal size after rectifying the difference, conversely, low external ion concentrations result in an influx of water and subsequent swelling of the cell before it returns to a normal size. (Fig. 2b). The model with each external ion concentration was confirmed to be stable. A model is stable when all possible starting states of the system return to a single, self-perpetuating point. For systems where the tonicity value is high or low, stability indicates that biologically the cell has rectified the differences in osmotic pressure inside and outside, and reached homoeostasis without a permanent swelling or shrinking (which could be damaging or fatal), as the osmolyte concentrations are altered to compensate for the change in extracellular environment. An example proof progression showing stability for the network in a hypertonic, and hypotonic state is shown in Supplementary Fig. 3.

Having shown the network functions in a biologically reasonable manner, we went on to study the model in the context of cell lines that can be validated, and for which gene expression data was available. We chose to study how cellular phenotypes pertaining to size, contractility, migration/membrane dynamics, attachment ability, and viability, are controlled and altered by membrane transport proteins and the solutes they transport.

**QN of membrane transport predicts stromal stimuli response**. Lymph nodes (LNs) are key immune hubs. Surveillance of lymph fluid by LNs for pathogens, and subsequent activation of a primary immune response is a key component of a functioning immune system. Tumour-derived fluid is generally collected

within the lymphatic system and drains to nearby LNs. The lymphatic system is exploited by many tumours during the process of metastasis, and the presence of tumour cells within the LN is a poor prognostic marker[33,34]. We have shown previously[35] that structural cells (fibroblastic reticular cells, (FRCs)), exposed to tumour-derived fluid, containing currently unknown tumour factors (TFs), but prior to metastasis to the LN, undergo

morphological and transcriptional changes, which are different in Early Tumour Draining (ETDLN) and late tumour draining (LTDLN) conditions. Among key changes to FRC function were genes associated with channel or transporter regulation. We used this dataset to guide an initial application of our model.

A specification describing changes in FRC morphology and phenotype in response to different experimental conditions was

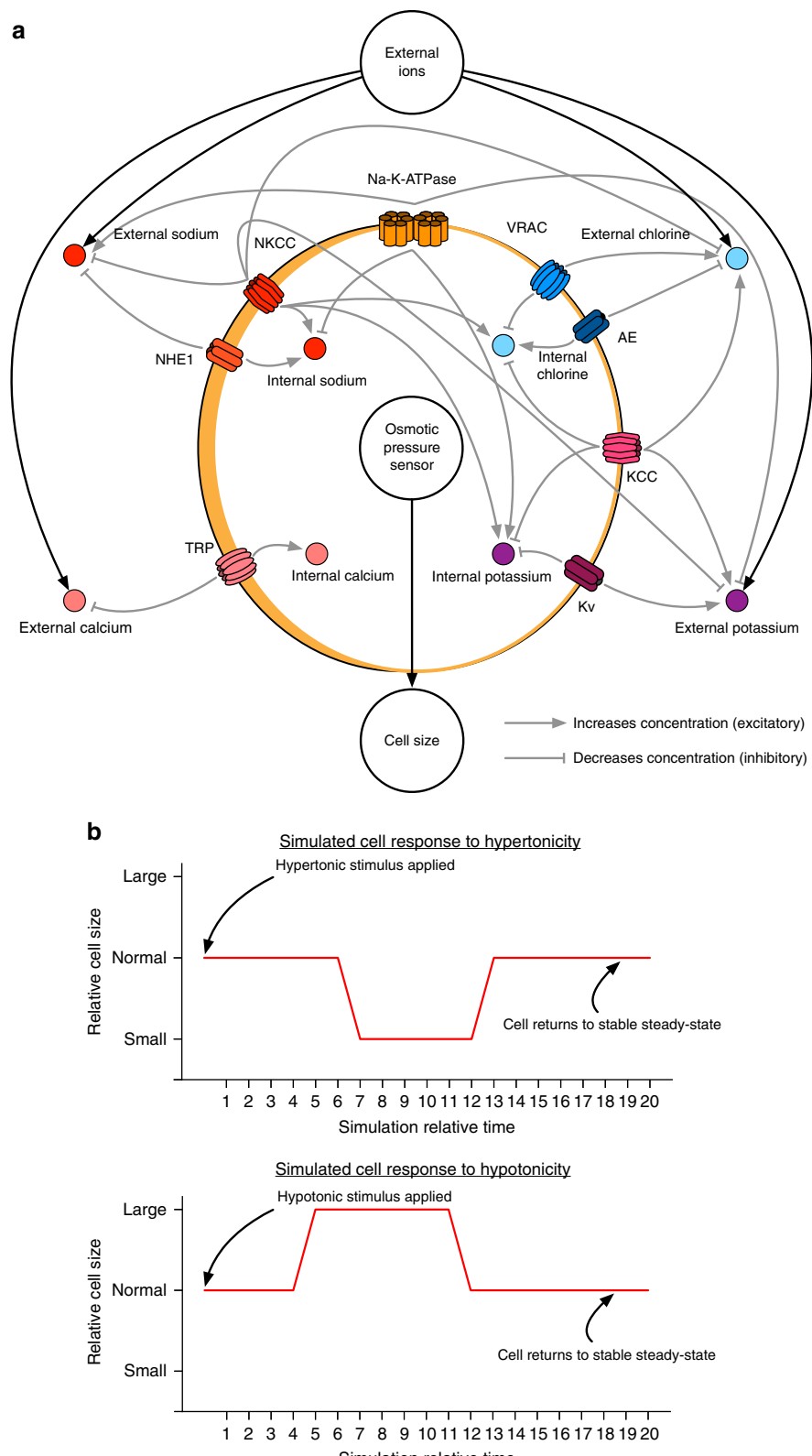

derived, focusing on cell viability, membrane dynamics/movement, attachment ability, contractility, and cell size (Table 3). The literature was also reviewed for the effects changes in ionic concentrations and membrane protein expression has on these phenotypes (Supplementary Data 3). Additionally, potentially influential, and significantly differently expressed proteins were collated from expression data (microarrays), and through verifying their importance in cell behaviour through the literature.

The initial model was then expanded to include these phenotypes, and the effects of ions and ion dependent signalling on them. A schematic of the expanded model is shown in Fig. 3a.

(The model in the BioModelAnalyzer interface is shown in Supplementary Fig. 4) We included domains for calcium signalling, pH maintenance, cellular metabolism/lactate transport, and the previously defined osmotic regulation network, though there is significant overlap between each. Additionally, nodes were added representing cellular phenotypes described in the specification, and linked to the model via the activity of ions, and ion dependent signalling cascades derived previously. Executable modelling also allows us to explicitly model complex phenomena, such as protein modification, lipid interactions, and cytoskeletal effects through altering the mathematical relationship between

**Table 3 Phenotype expression specification generated from gene arrays from FRCs**

| Cellular property | Expected change | Reasoning | Model expected results |
|---|---|---|---|
| *Viability* | | | |
| ETDLN | Increase | Experiments show increasing number of cells initially (Riedel et al.[35]). Gene array results also suggest proliferation pathways are upregulated | Viability increase from 2 to 3 or 4 in the presence of early stage TFs |
| LTDLN | No change/ Increase | Experiments show cell number increase plateus at later stage. Gene array results show proliferation pathways different to ETDLN. (Riedel et al.[35]) | Viability value of 2, or increase to 3 or 4 for late stage TFs |
| LPS | Increase | Proliferation helps swell the LN under immune conditions due to increasing number of FRCs. (Fletcher et al.[39]) | Viability increase from 2 to 3 or 4 in the presence of LPS |
| *Membrane dynamics/movement* | | | |
| ETDLN | Increase | Deregulation of membrane remodelling pathways in the gene array. (Riedel et al.[35]) | Migration increase from 2 to 3 or 4 in the presence of early stage TFs |
| LTDLN | Increase | Deregulation of membrane remodelling, as well as migration pathways in the gene array. (Riedel et al.[35]) | Migration increase from 2 to 3 or 4 in the presence of late stage TFs |
| LPS | Increase | Deregulation of movement/migration pathways, including MMPs in the gene array. (Malhotra et al.[36]) | Migration increase from 2 to 3 or 4 in the presence of LPS |
| *Attachment* | | | |
| ETDLN | Increase | Deregulation of junction molecules in gene array. (Riedel et al.[35]). Cell adhesive properties changed in attachment assay | Attachment increase from 2 to 3 or 4 in the presence of early stage TFs |
| LTDLN | Increase | Deregulation of junction molecules in gene array. (Riedel et al.[35]). Cell adhesive properties changed in attachment assay | Attachment increase from 2 to 3 or 4 in the presence of late stage TFs |
| LPS | Unknown | – | – |
| *Contractility* | | | |
| ETDLN | No change | Contractility assay shows increase only in LTDLN, microarray shows collagen and associated contraction genes deregulated at LTDLN only (Riedel et al.[35]) | Contractility remains at value of 2, in the presence of early stage TFs |
| LTDLN | Increase | Cell contractility assays show increase in ability to contract collagen gel. (Riedel et al.[35]) | Contractility increases from 2 to 3 or 4 in the presence of late stage TFs |
| LPS | Decrease | Contractility shown to be decreased under immune activation on LN. (Fletcher et al.[39], Astarita et al.[38]) | Contractility decreases from 2 to 1 or 0 in the presence of LPS |
| *Cell size* | | | |
| ETDLN | No change | Light scattering experiments show no change in cell size. (Riedel et al.[35]) | Cell size value of 1 in the presence of early stage TFs |
| LTDLN | No change | Light scattering experiments show no change in cell size. (Riedel et al.[35]) | Cell size value of 1 in the presence of late stage TFs |
| LPS | Increase | Light scattering experiments show increase in cell size. (Acton et al.[39]) | Cell size value of 2 in the presence of LPS |

Shown is the expected cellular behaviour under numerous conditions, the reasoning and proof, and the expected results from the model under different conditions (ETDLN, LTDLN, LPS)

**Fig. 2** Qualitative network model of osmotic regulation. **a** Schematic of qualitative network that responds to osmotic pressure. An abstract node for external ions is included, and changes within this node lead to pressure felt by the cell via the osmotic pressure sensor node. Channels involved in the appropriate osmotic response (influx of ions under high osmotic pressure, efflux under low osmotic pressure) are activated or upregulated resulting in an appropriate response to return pressure to normal. The model includes the core osmotic machinery, appropriate ions (sodium, potassium, chloride, and calcium), and abstract nodes for the overall control of external ions, pressure felt by the cell, and cell size. The total model contains 20 nodes. **b** Simulations of the network reveal the cell response to altered osmotic pressure. Simulations are allowed to stabilise, before osmotic pressure is perturbed through increasing or decreasing the external ions node. This leads to a cascade of events in which the osmotic pressure sensor node recognises the difference between external and internal ion concentrations, leading to a decrease (top) or increase (bottom) of cell size in response to hypertonicity (top) or hypotonicity (bottom), before channels are activated in order to rectify pressure changes, and the cell returns to a normal size

nodes, and adding nodes to account for different subregions or interactions in proteins. For example, Na-K-ATPase, which involves multiple subunits with different interactions with various regions of the protein. This represents an interface with structural biology where regions of the protein with structurally and functionally distinct differences are separated (Fig. 3b).

To explicitly adapt the model to tumour draining lymph node conditions, we focused on significantly deregulated probes within the previously published microarray that corresponded to ion channels or membrane transport, and had significant literature justification for impacting cellular phenotype (Supplementary Data). These genes were also built into the model. A node (named

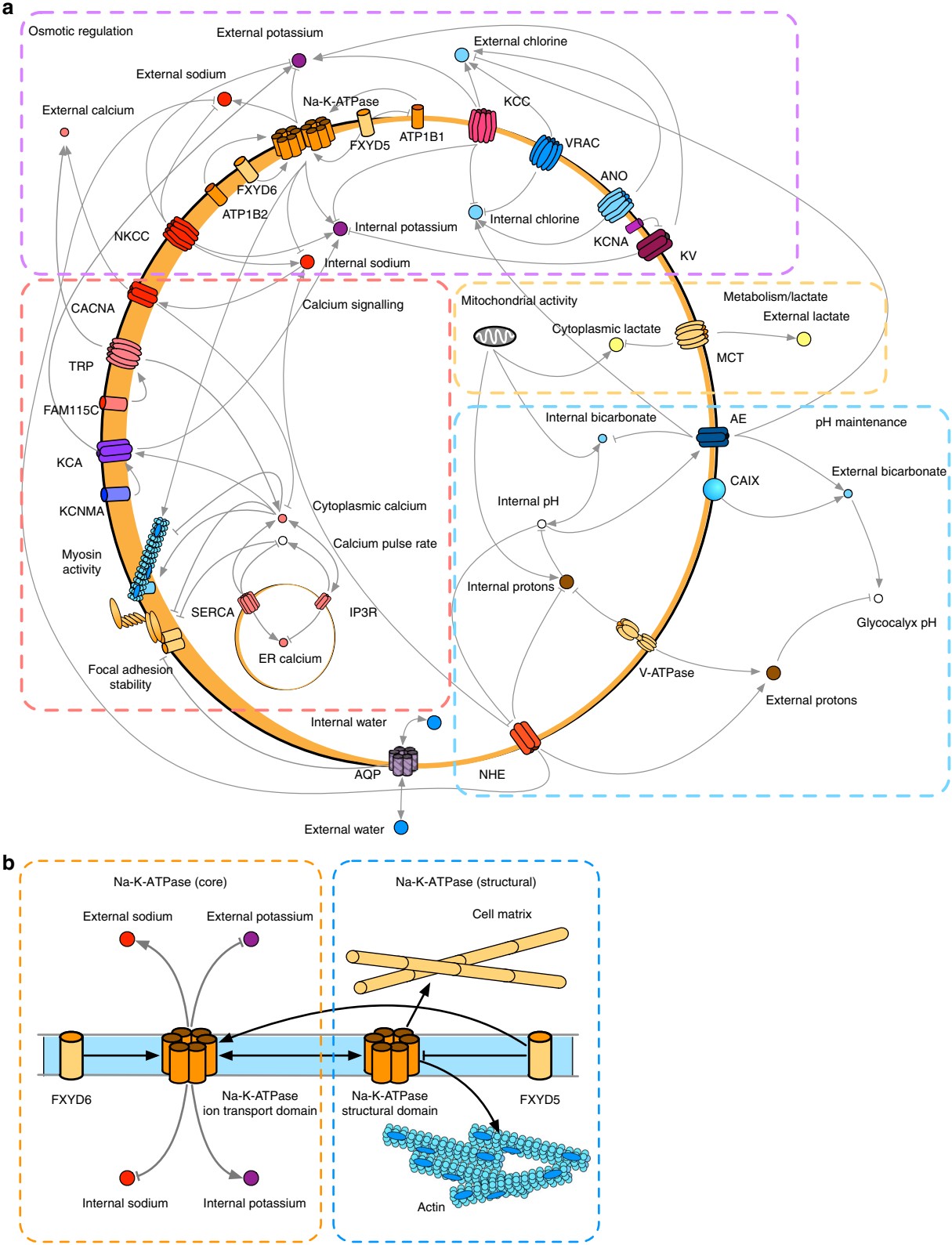

Tumour Factors) was added to the model to represent TDLN FRC conditions. Changes in the value of this node influence the respective probes from ETDLN or LTLDN to be upregulated or downregulated depending on their measured state in each condition.

When systemic tonicity was set to a value indicating no large difference to normal conditions (1), and tumour factors set to represent ETDLN conditions, stability analysis revealed nodes representing cell phenotypes stabilised at a global fixed-point matching the specification for ETDLN (Fig. 4b, i). In particular, we observed a coordinated change in ion channels that altered the activation state of many other nodes in the system. An example of coordinated changes caused by activation of a single channel is shown in Supplementary Fig. 5. As a result, cellular behaviours were activated or inhibited by the emergent behaviour of the model. Adjusting the Tumour Factors node to LTDLN conditions, revealed that cellular phenotype adapts again within the model due to the knock-on effects of changing the activity of a small number of ion channels. These results indicate that our model is able to explain the morphological changes in FRC cells exposed to TDLN, solely by taking into account expression changes of ion channels (Fig. 4b, ii).

To expand the model further, we chose to include data from previously reported studies of FRCs under immune activation. During an acute infection, LNs are the primary site of an initial immune response, during which they swell (caused by a loss of contraction of FRCs), but return to a resting state once the infection is resolved. This is contrary to tumours, in which the stimulus and phenotype changes on FRCs persists. Lipopolysaccharide (LPS) is a component of the outer membrane of gram-negative bacteria and is commonly used to induce an inflammatory response. We tested the predictive power of the model with the use of previously reported gene expression data for LN FRCs exposure to LPS[36].

We assessed the microarray for significantly deregulated membrane proteins and ion channels that had literature justification for an effect on cell phenotype (Fig. 4a, Supplementary Data 3). The changes in gene expression were then added to the model. We found that alteration of the six channels resulted in a stable state of the model in which viability, and movement/membrane dynamics increased, whereas contractility decreased. The full specification can be seen in Table 3. Interestingly, the model also predicts an increase in cell size, a phenotype confirmed by Acton et al.[37], and a phenotype that was not observed in FRCs exposed to TFs[35]. In particular, one study found that FRCs under immune stimuli reduce contractile capacity[38]—an inverse morphological change compared to TDLN FRCs. We found that the literature supported the phenotypes described in our model[36,39] (Fig. 4b, iii), without major refinement of the model parameters (minor refinement of the weights ascribed to each node was required).

It is suggested that a loss of contractility within FRCs during early immune challenge is a precursor to proliferative conduit expansion by FRCs[38,39], additionally both contractility assays and studies of the rigidity of TDLN suggest that the FRCs in TDLN are more contractile than their LPS counterparts[40]. Whilst CLEC2 driven activation of PDPN is known to be the driver of contractility changes in immune infection[38], the specific mechanism is unknown, the model suggests that alterations in calcium signalling, along with changes in regulation of cytoskeleton-associated channel domains can explain some of this difference.

To validate the causal links between ion channel expression and the phenotype of FRCs in vitro, we selected individual genes for knock-down experiments using siRNAs. Specifically, we investigated the effects of disrupting *ATP2A3* (SERCA3), *SLC9A1* (NHE1) and *FXYD5* (dysadherin) individually in FRCs, first confirming the knock-down efficiency by qRT-PCR (Supplementary Fig. 6A). We chose to focus on nodes that have a large effect on the phenotypes in the model, and involve sub-networks that are significantly changed upon addition of stimuli. The model must accurately predict not only the effect of the knockdown, but also the compensatory response of the network (upregulation of other proteins to counter ion imbalances, changes in pH maintenance performance etc) to be accurate. The model predicted that loss of *ATP2A3* would result in a reduction in attachment capacity, mainly through changes in calcium homoeostasis, and this was experimentally confirmed in vitro, where knockdown of *ATP2A3* reduced attachment as predicted (Fig. 4c). An example of the mechanistic prediction of cellular behaviour shift on knockdown of *ATP2A3* (SERCA3) is shown in Fig. 4d. In the QN, loss of *ATP2A3* leads to changes in the internal calcium compartmentalisation within the cell, and activation of calcium dependent channels. Resultant changes in focal adhesion and osmotic pressure leads to a reduction in cellular attachment ability. The model also predicted the involvement of *SLC9A1* (NHE1) in cell viability, and upregulation of this protein in LTDLN being a key reason for the subsequent increase in viability, due to changes in intracellular pH and potassium concentration. Knock-down of *SLC9A1* (NHE1) significantly reduced cell viability (Fig. 4e). We additionally performed a proliferation assay to explore the relationship between viability and proliferation (for discussion see Supplementary Note 2), and found that *SLC9A1* knockdown did not reduce the proliferation rate of FRCs, despite reducing their viability (Supplementary Fig. 6B)—a result that has been reported in the literature previously in breast cancer cells[41], and which confirms that our model is generally reproducing a viability phenotype. Finally, the model suggested that a reduction of *FXYD5* (dysadherin) would decrease cellular viability and increase attachment ability. Decreased *FXYD5* mRNA expression showed an increase in attachment properties and a small but significant reduction in cell viability (Fig. 4f), *FXYD5* knockdown also elicited a decrease in cellular proliferation rate (Supplementary Fig. 6C). Knockdown

---

**Fig. 3** Qualitative Network Model of the wider ion channel regulatory network. **a**, **b** Schematic of the network of ion channels involved in the model (**a**). Included are modules that involve osmotic regulation, calcium signalling, metabolism, and pH maintenance. Modules interact with each other heavily, and in particular osmotic regulation proteins are involved in many other modules. The schematic includes general trends of interactions, and does not show all nodes for completeness, but attempts to show the major protein types involved in each module, and the ionic behaviours considered. Individual nodes, however, can be much more complex than represented in the schematic, one example is shown in (**b**) of the Na-K-ATPase. In this case the Na-K-ATPase is split into two nodes, one representing the ion transport domain, and one representing the structural interactions of the domain. Both nodes interact with each other as they are part of the same protein, but subunits that associate in the protein complex elicit different effects. *FXYD6* influences the ion transport domain only, whereas *FXYD5* influences both the ion transport domain, and the structurally connected beta subunit domain, which interacts with the cytoskeleton and cell matrix independently of transport function. This separation of functional subdomains of the protein can also be used to incorporate phosphorylation states of the protein, where two nodes represent an unphosphorylated and phosphorylated state of the same protein

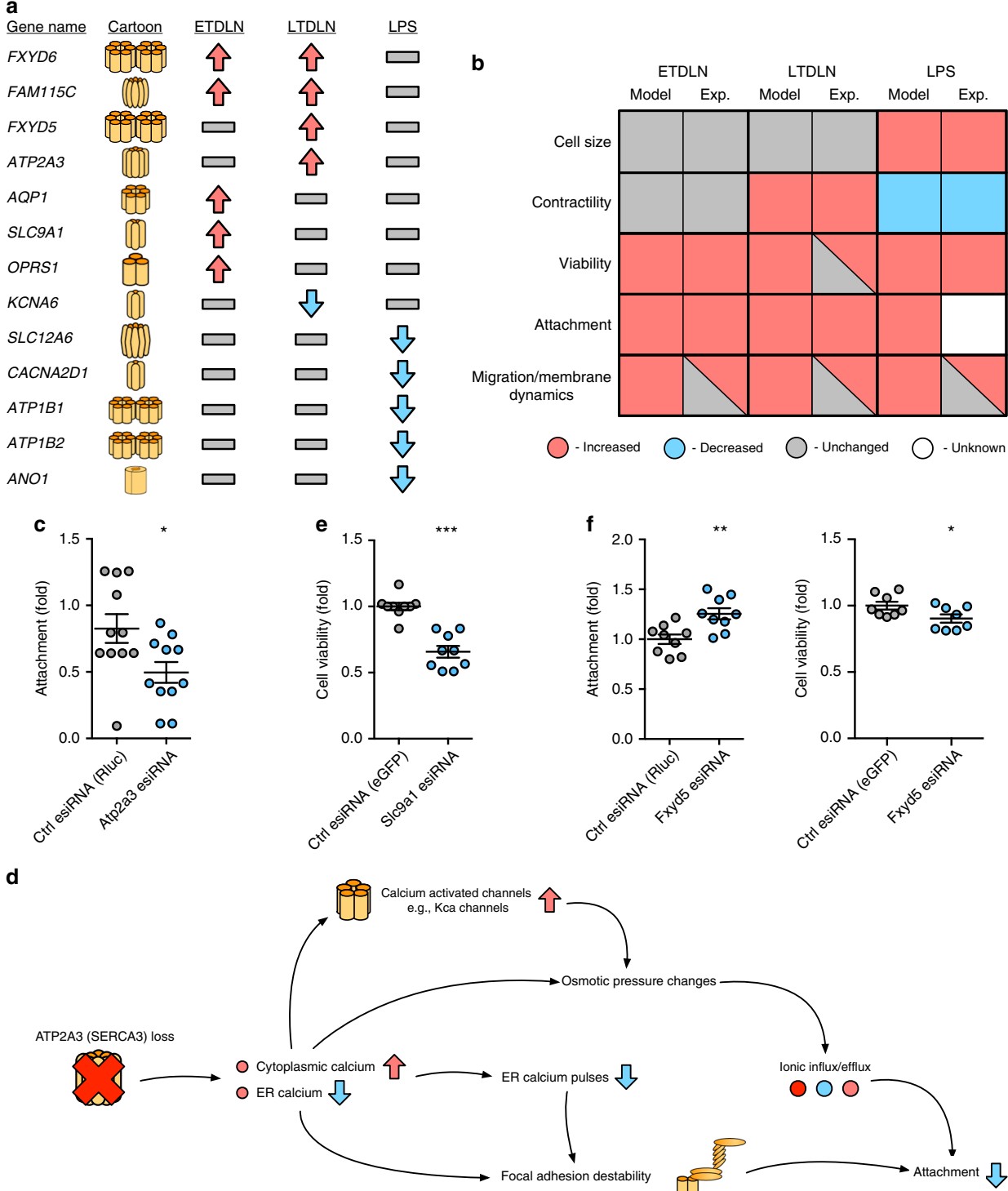

**Fig. 4** The qualitative network explains cellular phenotypes of Stromal cells in varying experimental conditions. Membrane proteins deregulated under varying stimuli, showing **a** Transport proteins deregulated in Early (ETDLN) and Late (LTDLN) stage after exposure to TFs, and after exposure to LPS. Proteins are upregulated (red arrows), downregulated (blue arrows), or unchanged (grey) in response to TFs or LPS. **b** Phenotype change for FRCs upon exposure to TFs for Early (ETDLN) and Late (LTDLN) stage. The Model output represents the physiological behaviour predicted by the model, and the experimental output represents the behaviour observed or implied at the cellular level from experiments or the gene array, and is independent of the model (see methods). Boxes containing two colours indicate phenotypes where there is contradictions/data is unclear. The model predicts an increase in viability, and a sustained increase in attachment and movement/membrane dynamics. Specific protein activity loss predictions are verified with experiments. **c**, **d** Knockdown of genes for (**c**) *ATP2A3*, and its effect on cellular attachment. The cascade predicted by the model to be underpinning this behaviour change is seen in part (**d**). **e**, **f** Also shown are siRNA knockdowns for *SLC9A1* and its effect on cell viability (**e**), and *FXYD5* and its effect on viability and attachment (**f**). Predicted mechanisms for these knockdowns are included in Supplementary Fig. 6. *$P < 0.05$, **$P < 0.01$, ***$P < 0.001$ using two-tailed unpaired *t*-test. Error bars represent standard deviation. Shown are 11 replicates (**c**), and 9 replicates (**e**, **f**)

**Table 4 Phenotype expression specification generated from gene arrays from MEFs**

| Cellular property | Expected change | Reasoning | Model expected results |
|---|---|---|---|
| *Viability* | | | |
| HOM vs. HET | No change | No increase in proliferative ability or apoptosis observed (Kerr et al.[42]) | Viability value of 2 under HOM conditions |
| *Attachment* | | | |
| HOM vs. HET | Decrease | Increased colonisation phenotype, more invasive cells (Kerr et al.[42]) | Attachment decrease from 2 to 1 or 0 under HOM conditions |
| *Cell size* | | | |
| HOM vs. HET | Increase | No observed change in cell size (Kerr et al.[42]) | Cell size value of 1 under HOM conditions |

Shown is the expected cellular behaviour under homozygous *Kras* mutant conditions when compared to heterozygous conditions, the reasoning, and the expected results from the model

cascades from the model for *SLC9A1*, and *FXYD5* are also shown in Supplementary Fig. 6D and E. Together, we confirm the validity of the model by isolated knock-down experiments of single components and measuring their predicted effects on cellular behaviour.

**Murine embryonic fibroblast QN predicts attachment behaviour.** Due to the ubiquity of the regulatory networks involved in osmotic regulation, we further tested the model on cells from a different system. We employed Mouse Embryonic Fibroblasts (MEFs) that have been previously characterised[42]. MEFs isolated from $p53^{-/-};Kras^{G12D}$ embryos exhibit different phenotypes depending on whether they are heterozygous ($Kras^{G12D/+}$; hereafter termed HET) or homozygous ($Kras^{G12D/G12D}$; HOM) for the $Kras^{G12D}$ mutant allele. There is no observed proliferative difference, or cell size difference between the two genotypes, but HOM MEFs exhibit a highly penetrant colonisation phenotype, implying reduced attachment capabilities (Table 4).

We again searched for ion channels and membrane proteins within the microarray[42] that were significantly deregulated and had significant literature justification for an effect on cell phenotype (Fig. 5a, Supplementary Data 4). We then incorporated these changes into the QN and determined that our model also supports the behavioural changes recorded in HET and HOM MEFs[42]. Indeed, when the QN was adjusted to reflect the gene expression from HOM MEFs compared to HET (Fig. 5b) the model predicts that viability remained unchanged, attachment was decreased, and cell size remains unchanged.

**Channel specific inhibitors allow control of phenotype.** By exploring the consequences of knockouts on the model, we find that upregulation of *SLC4A3* (AE chloride/bicarbonate transporter) is linked to viability, and impacts cellular attachment. We applied the *SLC4* selective inhibitor 4,4′-diisothiocyanatostilbene-2,2′-disulfonic acid (DIDS) to HOM MEFs in vitro, to abolish *SLC4* activity, and measured resultant changes in cell viability. We show that elimination of *SLC4* activity in a DIDs dependent manner leads to a marked reduction in cell viability (Fig. 5c, i).

Going further, it is known that selective inhibition of *SLC9* (NHE sodium/proton transporters) by amiloride related compounds reduces cellular viability, particularly in cancer systems[43,44], consistent with our siRNA knockdown experiments in LN FRCs. NHE transporters, primarily NHE1 mediate the exchange of protons for sodium across the plasma membrane, and are a key component of pH maintenance in mammalian cells. NHE1 upregulation is suggested to be the key factor leading to tumour metastasis in breast cancer[41]. Additionally, NHE1 is a promising target for selective anti-cancer therapeutics, by targeting the aberrant pH present in all cancer cells[43,45,46]. Translation of *SLC9* inhibitors to in vivo systems however, has

initially been unsuccessful[47], in part because cells were still able to metastasise to regions dense in extracellular matrix (ECM). Our model predicts that by combining the effects of the *SLC4* inhibitor DIDs with *SLC9* inhibitors such as amiloride should result in (i) decreased viability due to the disruption of both intracellular chloride and sodium concentrations, and of the pH maintenance ability of the cells, and (ii) a significant reduction in cellular attachment properties (we note that attachment is already low in our model under HOM conditions, but loss of *SLC4* and *SLC9* leads to a further decrease).

We show experimentally that addition of *SLC9* inhibitor 5-(N-ethyl-N-isopropyl)amiloride (EIPA) to HOM MEFs in vitro also leads to a reduction in cell viability (Fig. 5c, ii), an effect that is retained when the drugs are applied in combination, though Bliss Independence analysis indicates that this is likely a purely additive effect. Additionally, applying DIDs and EIPA and studying attachment ability of cells (Fig. 5d), reveals a potentially synergistic reduction in attachment capability of MEFs. Biological repeats are shown in Supplementary Fig. 7A, confirming initial results, channel blocking was confirmed with the use of a chloride specific dye MQAE (Supplementary Fig. 7B).

This is wholly consistent with mouse models of mastocitomas where this pair of drugs were proposed to have reduced metastasis[48]. Thus we suggest that administration of *SLC9* specific inhibitors in concert with compounds that inhibit cellular attachment such as DIDs could overcome past issues with the use of anti-*SLC9* drug regimes in some cancers.

Additionally, we chose to study the effect of the drug amiloride in combination with DIDs. Amiloride is not a specific *SLC9* inhibitor, but generally inhibits sodium channels. Amiloride is already used to treat hypertension or hypokalemia, and thus represents a more immediately clinically relevant *SLC9* inhibitor. We show that addition of amiloirde hydrochloride (AHCL) with DIDs results in the same significant decrease in viability (Fig. 5e), and synergistic decrease in attachment (Fig. 5f) that are observed with EIPA and DIDs.

**Discussion**

The role of ion channels in cellular transformation in disease has been discussed previously[15,27,49]. Many diverse cellular phenotypes are known to rely heavily on the flux of ions and solutes, which are controlled by the changes in activity of membrane transporters[50–52]. We highlight the importance of ion channels in cancer with a data science approach, using the large TCGA database as a source for classifying cancers based on channel expression only. We also highlight the use of classification feature weights for focusing on particular genes and gene subtypes. Aquaporins for example consistently appear in the analysis, and are involved in the model in the later part of the paper, yet their impact on cancer and greater regulatory control is not well

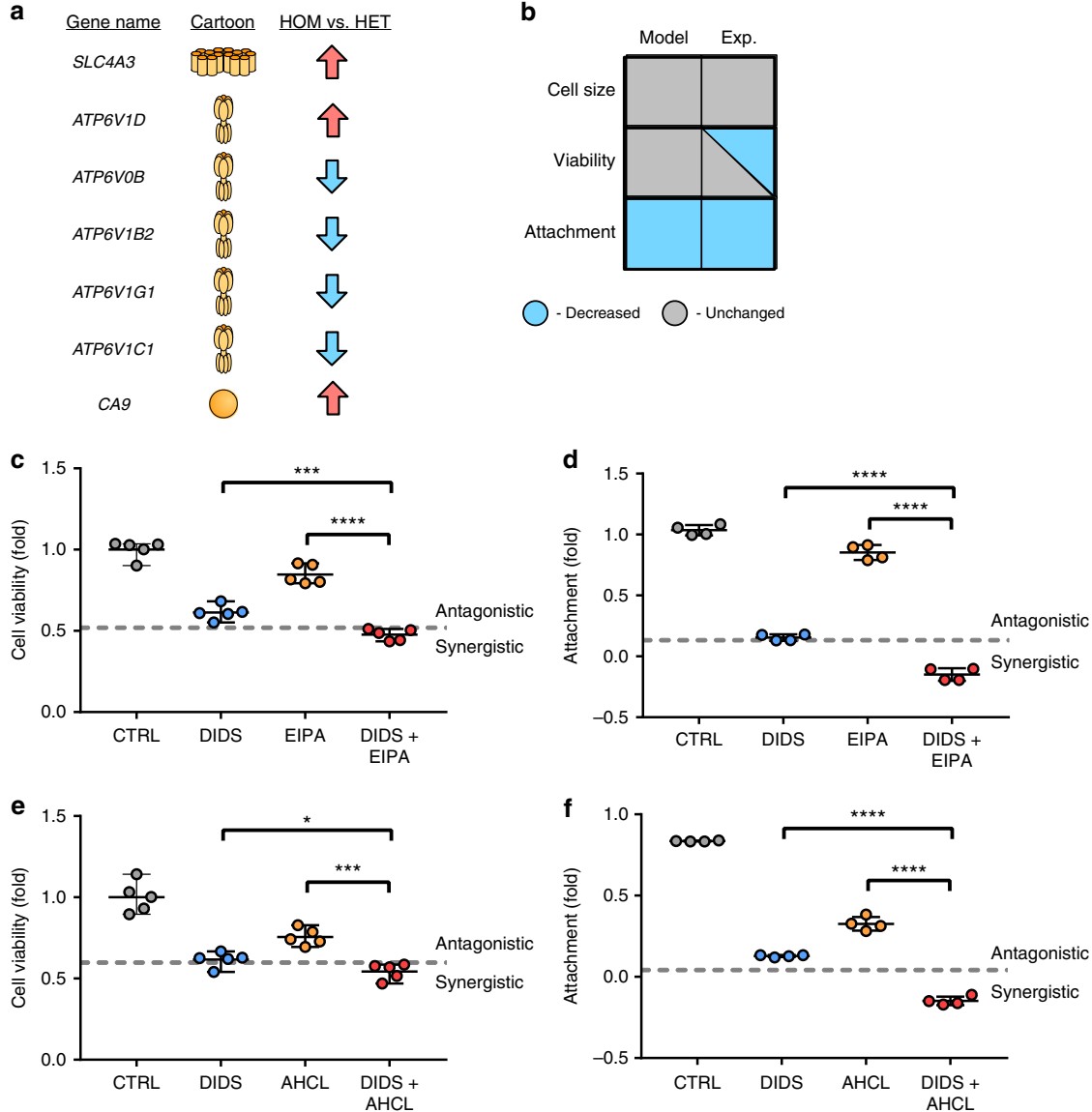

**Fig. 5** Application of the qualitative network to murine embryonic fibroblasts. Expansion of the QN to p53$^{-/-}$, Kras$^{G12D}$/+ (HET) and p53$^{-/-}$, Kras$^{G12D/G12D}$ (HOM) MEFs. **a** Transport proteins deregulated within HOM MEFs when compared to HET MEFs. Proteins are upregulated (red arrows), downregulated (blue arrows), or unchanged (grey). **b** Phenotype change for MEFS when HOM MEFs are compared to HET MEFs. The model output represents the physiological behaviour predicted by the model when proteins from (**a**) are deregulated in the model in the same manner as in the gene array. Experimental phenotype represents behaviour reported previously[42], where it is known. The model predicts that attachment will be significantly different between the two cell types, but viability and cell size will remain unchanged. **c** Effects of application of channel inhibitors to HOM MEFs in vitro showing application of 10 μM DIDs for 72 h, resulting in decreased cellular attachment (left), and application of 10 nM EIPA ± 10 μM DIDs (right). Data are technical replicates. Dotted lines represent calculated Bliss independence values. **d** Effects of application of 10 μM DIDs ± 10 nM EIPA on viability in HOM MEFs. **e** Effects of application of 10 μM DIDs ± 10 nM AHCL on attachment in HOM MEFs. **f** Effects of application of 10 μM DIDs ± 10 nM AHCL on viability in HOM MEFs. *P < 0.05, ***P < 0.001, using two tailed unpaired t-test. Error bars represent standard deviation. Shown are 5 replicates (**c**), and 4 replicates (**d**–**f**)

understood, and a more detailed executable model may in the future lead to further insights.

Here, we describe the first computational model of an interlinked network of ion channels and transporters involved in osmotic regulation, and show that dysregulation of the membrane transport network may explain and impact the behaviour of cells in disease such as cancer. Whilst many recent studies have highlighted membrane transporters as markers for cancers, little attempt has been made at a mechanistic explanation for the transformation of cells in cancer with respect to membrane transport. We show that membrane transporters not only correlate specifically with different cellular phenotypes associated

with cancer, but that they are a principal agent in the transformation of cellular phenotype under many conditions. Furthermore, our data supports the notion that ion channels are a highly effective therapeutic target in the treatment of transformative disease, and demonstrate this with experimental examples of targeted changes in cellular phenotype using SiRNA knockdown experiments, and channel inhibitors. We also provide an explanation and potential solution to the failure of *SLC9* specific inhibitors in the treatment of cancer metastasis, and confirm current studies[53–55] suggesting that dysadherin is a potential therapeutic target for cancer associated cellular transformations.

Lymph node transformation upon exposure to tumour factors is still a poorly understood process. It is accepted that LNs respond to tumour factors before the arrival of cancer cells, and that they form a pre-metastatic environment suitable for the spreading and embedding of cancers[35]. Stromal cells are highly integral to this process, and whilst the physiological and phenotypical changes associated with this process have been demonstrated, a mechanism has not previously been elucidated. We propose that the reorganisation of ion channel expression is one of the key mechanisms by which cellular transformation of TDLN FRCs occurs. We have shown that targeted knockdown of certain channels in vitro systems can modulate behaviour in specific ways within FRCs.

Additionally, the model can be rapidly adapted to different challenges with simple adjustments to the base model directed by readily available single-cell expression data. LPS exposure represents changes associated with the LN immune response, and is an entirely different physiological stimulus to TFs. We find that our model accurately predicted the cellular responses displayed in the literature, without major iteration of the mathematical underpinning of the model. The fact that ion channel expression can describe cellular behaviours such as attachment capability is not surprising—the impact of many individual ion channels on cellular behaviour has been extensively studied, however, our QN represents the first case of an attempt of unifying their concerted actions on cellular phenotype mechanistically.

We also apply the model to a different cell system—that of MEFs heterozygous for a *Kras* mutant gene, and their homozygous mutant counterparts[42], that show a distinct phenotype. We show that ion channel dysregulation correlates with the observed cell phenotype and that this behaviour can be manipulated with the inhibition of specific channels, and thus show that physiologically distinct systems alter their behaviour using a common apparatus.

We go on to prove a predicted synergistic effect on attachment, and an additive effect on viability in MEFs that has been hinted at previously[56]. We suggest that the use of inhibitors of both *SLC4* and *SLC9* channels simultaneously will overcome previous limitations on the treatment of metastatic cancers using *SLC9* inhibitors alone[47], and demonstrate that this is feasible with in vitro experiments. Finally, the MEF data raise an important issue regarding the clinical interpretation of cellular cancer phenotypes in different contexts. In mastocytoma models loss of attachment ability is linked with reduced metastasis[48]. In contrast the decreased attachment of HOM MEFs in vitro correlates with increased metastasic potential[42]. The QN presents a platform for targeted drugability of disease states, particularly poignant in this case—as there are many highly specific, naturally occurring inhibitors of ion channels derived from natural compounds such as venoms and neurotoxins readily available[57–59].

Whilst we do not claim to completely encapsulate all cellular behaviour, or the only mechanism for its change, we have shown that focusing on ion channels is enough to partly explain cellular morphological change, and provide useful predictions for modulation of cellular behaviour, though we note that predictions validated here are only tested in vitro.

In summary, using data science, and an executable model tuned on experimental gene expression data, we demonstrate that ion channel expression correlates with, and coordinates cellular transformation in disease states, notably cancer. We show that expression of channel allows discrimination of cancer types, and that regulators of osmosis are key channels involved in the discrimination of cancer and non-cancer cells. We generate an executable model to explore the osmotic regulation network. We validate our model with knockdown experiments and inhibitor compounds, and show that targeted disruption of this interconnected system of transporters in vitro can alter cellular behaviour in predictable ways. Further work will refine components of the model to be more detailed, such as calcium signalling and aquaporin regulation, and expand the model to more clinically relevant systems.

## Methods

**Analysis of TCGA data on membrane transport proteins**. The RNA-seq counts data was downloaded from the TCGA data portal (https://portal.gdc.cancer.gov/), and a CPM (counts per millions) normalisation was applied on all samples.

The resultant dataset contains 11,574 samples, 10,831 (93.6%) of which are cancer cells, 743 (6.4%) of which are non-cancer cells, from 37 different projects covering a range of cancers.

t-SNE was run using the t-SNE method available as part of the sklearn package in python. In order to speed up the running of the t-SNE algorithm, principal component analysis (PCA) was initially applied to the data to reduce it down to 50 dimensions. We implemented PCA using the sklearnPCA module. The reduced dimension data was then used as input for t-SNE clustering. We chose to implement the barnes-hut methodology, with a perplexity of 15, learning rate of 1000, and using 1000 iterations of the algorithm. We implemented the t-SNE module within sklearn.

Once t-SNE clustering was performed, we further clustered the resultant 2-dimesional map using k-means clustering from the KMeans module within sklearn, with a target number of clusters of 16 to generate the central points of each cluster. Central points of each cluster were then used to generate the voronoi surface using the Voronoi module within scipy in python. Figures were generated with Seaborn within Matplotlib v2.0.2[60].

We chose to utilise the XGboost module within sklearn to perform binary classification[61]. The gene expression data was given an extra classifier based on whether it is reported to come from a tumour sample (value of 1), or non-tumour sample (value of 0). The dataset was divided into a test set consisting of a random sample of 20% of the data. The remaining 80% of the data was used to proceed with learning, during which the sample type classifier (tumour vs. normal) was removed. Parameter grid searching revealed the optimal parameters for this dataset to be a maximum tree depth of 7, number of estimators to include as 400, and the learning rate to be 0.1. We then trained the module using K-folds cross validation on 10% subsets of the data, iterating with the F1 score. Early stopping was used after 40 rounds to prevent overfitting. The resultant model was then used to predict the classification of the previously unseen test set. Model accuracy was assessed using the Matthews Correlation Coefficient. Feature weights and feature gains (Supplementary Fig. 1D) show the more commonly used genes for classification, and strongest determiners of cell type, respectively.

Heatmaps of feature weights were generated by taking random subsets of 25 cancer, and 25 non-cancer samples. Only the top 25 feature weighted genes were then used for heatmaps. Heatmaps were generated and clustered using hierarchical clustering. The classifier column (value of 1 for cancer samples, and 0 for non-cancer) was included in the clustering to ensure separation of the two sample types. Values were normalised before being plotted using Seaborn and Matplotlib[60].

**Qualitative network models**. Qualitative networks are an extension of Boolean networks, described in detail in[62]. Briefly, a qualitative network, Q(V,T,N), is formally defined as consisting of variables ($V = (\nu_1, \nu_2, \nu_3, \ldots \nu_n)$, representing genes, proteins, or other components), target functions (T) and granularity ($N+1$). A state of the system is a finite map $s : V \rightarrow \{0 \ldots N\}$. Each variable has a specific target function associated with it, and variables update synchronously. The target functions determine the update of the network from a state $s = (d_1, d_2, d_3, \ldots d_n)$ to the next state $s' = (d'_1, d'_2, d'_3, \ldots d'_n)$ computed as in Eq. 1:

$$d'_i = \begin{cases} d_i + 1 & d_i < N \text{ and } d_i < T_i(s) \\ d_i - 1 & d_i - 1 \, d_i > 0 \text{ and } d_i > T_i(s) \\ d_i & \text{otherwise} \end{cases} \qquad (1)$$

As the model state space is finite, the set of recurring states (states can reach themselves after a finite number of steps) is never empty. A QN is said to be stabilising if there exists a unique recurring state and the next state $s' = s$.

In biological terms values in these models represent states of biological activity, such as the ranges of the expression of proteins, varying concentrations of chemicals within systems. The associated target function is an algebraic formula that describes how the variable responds to upstream variables; that is to say, how it is activated and deactivated in response to stimuli.

Stability refers to the concept by which a model can eventually reach a point in simulation from all possible starting states where it only updates to the state it is already in—it reaches a single fixed point. Stability in terms of biology therefore can be used as a specification for such biological phenomena as homoeostasis or equilibria.

**The BioModel Analyzer**. The BioModelAnalyzer (BMA) is a graphical interface for the construction and analysis of qualitative networks, is publicly available (http://biomodelanalyzer.org) and is described in detail elsewhere[63,64]. It allows construction of QNs through a simple drag-and-drop interface, and testing and simulation can be performed through the use of in-built stability, LTL analysis (bounded model checking) and simulation functions. All models presented in this study are available in their entirety at https://doi.org/10.5281/zenodo.1257326.

**Details of specification for TDLNs.** We recently demonstrated[35] that LN FRCs undergo transcriptional and phenotypic changes when exposed to tumour factors (TFs). In particular, we observed substantial remodelling within the conduit network, of which FRCs are the key structural component, over time. Conduit diameters increased, the number of branches within them decreased, and conduits appear to become more contractile. We were able to surmise the changes in cell phenotype into a group of specific cell behaviours that are different between control non-tumour draining lymph nodes (NTDLN), and LN in the early and late stages of exposure to tumour factors named early tumour draining lymph nodes (ETDLN), and late tumour draining lymph nodes (LTDLN) respectively. Findings by Reidel et al. indicate that there is a difference between the phenotypes of FRCs following 4 days of exposure to TFs (ETDLN), and 11 days (LTDLN), and we chose to use these time points as our definition of early and late stage characteristics.

In TDLNs, proliferative expansion of FRCs contributed to the remodelling observed. During this transformation process, as would be expected during large structural reorganisation, there was a significant dysregulation to cell–cell adhesion pathways upon exposure to TFs, resulting in higher cell–cell attachment. Moreover, FRCs exposed to TF become more contractile, and the conduit network is known to be more elastic in malignant systems[40,65,66] (Supplementary Fig. 8A). In contrast, in the case of inflammation, FRCs exposed to LPS exhibit a reduction in contractility with increased proliferation[38,39].

Beyond morphological changes, gene array analysis confirmed that proliferation and cellular junction/attachment pathways were deregulated at the transcriptional level after exposure (Supplementary Fig. 8B), and also highlighted movement/migration pathways as undergoing significant alteration. Whilst it is unclear whether FRCs in the conduit network move substantially during transformation there is a large degree of overlap between cell motility and cell membrane dynamics[67,68]. As such we chose to interpret and model movement/migration pathway deregulation in the context of dynamic motion of the membrane. The membrane dynamics of FRC networks are likely critical to their immune function, regulating immune cell access to antigen-rich lymph fluid transported within conduits, from which they sample lymph directly[69,70]. On this basis, we chose to have a single output node for movement/membrane dynamics together, as these behaviours are difficult to dissociate, and are highly interrelated. By combining morphological and gene array analysis we are able to generate a description of cellular behaviour under different conditions (Supplementary Fig. 8C).

**Details of specification for LPS exposed LNs.** The presence of LPS in the LN is a sign of immune infection. Upon detection of a bacterial or other immune threats, the LN initiates an immune reaction resulting in morphological changes to the organ itself, and to the cells that form it. Stromal cells in particular, such as FRCs, transform significantly under immune activation of the LN[39]. FRCs are observed to reduce in their contractile properties, allowing an initial expansion of the conduit network of which they are the principal structural element[38]. The proliferative rate of FRCs under immune reaction also increases, contributing to LN swelling[36,39]. Gene pathway deregulation, and integrin expression predict that migration/motile ability of FRCs under immune conditions will be increased[36,71], a position supported by general understanding of organ morphology changes[39].

**Details of specification for Kras$^{G12D}$ MEFs.** A specification for the phenotypic differences between $p53^{-/-};Kras^{G12D/+}$ (HET) and $p53^{-/-};Kras^{G12D/G12D}$ (HOM) MEFs was generated through reference to published data[42]. Cell size and proliferative capabilities remain unchanged, but metastatic potential through a colonisation assay increased in HOMs.

**Details of model generation.** The model is generated form the literature. Interactions between proteins and ions are determined through thorough literature review, and refined against emergent behaviour of the model, literature justifications for biological function of genes involved in the model are detailed in Supplementary Data 5. The model is broken down into components for an easier understanding hereafter.

The osmotic regulation model was constructed based on the canonical proteins involved in primary osmotic response. Osmotic pressure changes are sensed by a cell, and channels involved in the response are activated, or in some cases upregulated. The activation of these channels results in a rapid influx or efflux of ions down their concentration gradients. As the osmotic pressure gradient is rectified, then the channels reduce in activity, and osmotic pressure is maintained. As the mechanisms by which cells sense and response directly to osmotic pressure are complex, and not fully understood, we chose to abstract the osmotic pressure sensed by the cell into a single node that corresponds to the difference between internal and external osmolyte concentrations. We increased the granularity of

nodes representation transporting proteins, as this allowed the network to stabilise despite its cyclical nature. Biologically this slowness (as nodes can only update by a single step at a time) to respond to changes in osmotic pressure represents both the upregulation of osmotic response genes, and the graded response mechanosensitive channels can have to membrane pressure.

A module was included representing calcium signalling. The temporal and spatial properties of calcium exert numerous controls over cellular behaviours including movement and migration, proliferation, and adhesion. Calcium is also somewhat involved in the osmotic response. The calcium signalling region of the model contains pumps involved in moving calcium between the plasma membrane, cytoplasm, and endoplasmic reticulum (ER), as well as an abstracted node for calcium pulse rate. Whilst we are unable to capture spatial polarisation properties explicitly within our model (though this is theoretically possible), the rate of calcium pulses from the ER is directly linked to cellular behaviour such as focal adhesion stability. We chose to represent calcium pulses with an integer value, rather than a cycle within the model, in order to make stability analysis possible.

Maintenance of pH within mammalian cells is performed via a mechanism that is similar to osmotic regulation, and many components are shared, as such we explicitly included a pH maintenance motif. The pH of cellular components must be maintained within a standard range, in order to maintain enzymatic function, and metabolic rates. pH dependent proteins change their activity levels in response to different pHs, resulting in a robust equilibrium. We chose to include the pH maintenance network due to the overlap of many proteins with those in the osmoregulatory network, such as NHE proteins, and due to the integral role pH plays in cellular behaviour, and particularly cancer phenotype. We chose to include pH domains for the glycocalyx (the sugar containing shell surrounding a cell), and the cytoplasm. We increased the granularity of the pH node from 4 (in the majority of the rest of the model) to 6. This allowed a better discrimination of the effects of certain protein knockdowns on pH (as pH could be altered by a small amount that is significant but not registered when the node is 0–4).

We chose not to explicitly represent any metabolic processes due to the complexity of the model. Instead we chose to represent metabolic output in terms of an explicit mitochondrial activity node. This node influences the available concentrations of metabolic outputs, such as protons and lactate. This node is altered in disease states where metabolism is disrupted. This is the case in HOM cells[42].

By curation of the literature, phenotypes and their associations with ion concentrations within regions of the cell were generated. These were then added to the model, phenotypes involved, and associated references are included in Supplementary Data 3.

**Verification of the model.** Most conditions were tested using BioModelAnalyzer using the unbounded stability proof algorithm from Cook et al.[63], through the BioModelAnalyzer web interface. For a single condition (External Ion Concentration of 2, with no other external factors present), unbounded model checking was not possible, and bounded checking performed through the command line version of the tool[18] were tested, using pairs of queries (eventually selfloop) and (eventually selfloop and not A), where A is the selfloop state of the system identified by the first query. The bound for simulations was set to 20 steps. Testing with larger bounds timed out on our workstations. (Time limit 24 h, workstation with Intel xeon CPU E5-2697 v3 @ 2.60 GHz processor).

**Analysis of microarrays.** TDLN microarray data was normalised and pre-processed using the lumi package. In short, the bottom 50% of the genes by inter-sample variance were removed, variance stabilisation transformation was performed followed by quantile normalisation. Genes were then filtered by $P$-value ($P > 0.05$ were excluded), and fold change ±1.5.

LPS exposed microarray data was analysed using GenePattern, specifically the ComparativeMarkerSelection tool was used. After normalisation using the PreprocessDataset module, resultant data was fed into comparative marker selection for analysis. A two-sided $t$-test was used to generate $P$-values, and the results were filtered by $P$-value ($P > 0.05$ were excluded), and fold change ±1.5.

MEF microarray data was normalised and preprocessed using the lumi package. In short, the bottom 50% of the genes by inter-sample variance were removed, variance stabilisation transformation was performed followed by quantile normalisation. Genes were then filtered by $P$-value ($P > 0.05$ were excluded), and fold change (genes ± 1.3 were included).

**Cell culture.** Primary murine fibroblastic reticular cells (pmFRCs) were isolated from murine lymph nodes of C57BL/6 mice. Lymph nodes were mechanically disrupted and digested in a 500 μl mixture of 1 mg/ml collagenase A (Roche) and 0.4 mg/ml DNase I (Roche) in PBS at 37 °C for 30 min with 600 r.p.m. rotation. Following centrifugation at 1000 r.p.m. for 5 min, the supernatant was discarded and replaced with 500 μl of PBS containing 1 mg/ml Collagenase D (Roche) and 0.4 mg/ml DNase I. The mixture returned to 37 °C for 20 min with 600 rpm rotation before addition of EDTA (final concentration 10 mM). PmFRCs were characterised based on their expression of Pdpn and VCAM-1 and their lack of expression of CD45 and CD31 via flow cytometry. Cells were maintained in RPMI (R875, Sigma Aldrich) with 10% foetal bovine serum (Sigma Aldrich), 10 mM

HEPES, 100 U/ml penicillin-streptomycin (both Life Technologies), 15 μM beta-mercaptoethanol (Sigma-Aldrich) in a 37 °C incubator with 5% $CO_2$.

For MEFs, animals were maintained under SPF conditions and in compliance with UK Home Office regulations. $Kras^{LSL-G12D}$ mice were crossbred to p53$^{Fx}$ to obtain mixed background (C57Bl/6/129/Sv) $Kras^{LSL-G12D/+}$;p53$^{Fx/Fx}$ mice. For MEF generation, $Kras^{LSL-G12D/+}$;p53$^{Fx/Fx}$ mice were crossed, embryos collected at embryonic day (E)12.5 and fibroblasts isolated. Adenoviral cre-mediated recombination of alleles was performed one passage after isolation using $5 \times 10^7$ plaque-forming units per $1 \times 10^6$ cells (University of Iowa, Viral Vector Core). Cells were cultured in DMEM supplemented with 10% FBS, 2 mM L-glutamine (Life Technologies) and maintained at 37 °C with 5% $CO_2$.

All cells were regularly checked for mycoplasma contamination.

**RNA interference**. Commercially available endoribonuclease-prepared siRNAs pools comprising a mixture of siRNAs that all target the same mRNA sequence (Sigma) were used for the targeted knock-down of the mRNAs for *ATP2A3*, *FXYD5* and *SLC9A1*. Briefly, 100,000 pmFRCs were reversely transfected with 1600 ng esiRNA with the help of Lipofectamine RNAiMAX (Thermo Fisher Scientific) transfection reagent in a six-well plate. Twenty-four hours later the medium was exchanged and 72 h after assays (attachment and qRT-PCR) were performed.

**Viability assays**. 1000 Murine primary pmFRCs were reversely transfected with 100 ng esiRNA (Sigma) and seeded in triplicates into a 96-well plate. Twenty-four hours later the medium was exchanged and 72 h later cell viability was measured by the addition of CellTiter-Blue (Promega) viability assay reagent. After 6 h of incubation under normal growth conditions, fluorescence intensities were measured at 560$_{EX}$ nm/590$_{EM}$ nm with a microplate reader (Tecan).

Kras$^{G12D/G12D}$; p53$^{null}$ MEFs were seeded at 3000 cells per well in clear 96 well plates and allowed to adhere overnight. Cells were then treated with indicated concentrations of DIDS, EIPA, AHCL or a combination of these and cell viability was measured by 3-(4,5-dimethylthiazol-2-yl)-2,5-diphenyltetrazolium bromide (MTT, Sigma) assay after 72 h. Cells were incubated with 0.5 mg/ml MTT solution for 3 h at 37 °C before stain was resuspended in DMSO and absorbance at 570 nm determined by microplate reader (Tecan).

**Proliferation assays**. 5000 Murine primary pmFRCs were reversely transfected with 160 ng esiRNA (Sigma) and seeded in quadruplicate into a 96-well plate. Media was exchanged 6 h post-transfection. Proliferation was determined by measuring cellular confluence, over 70 h, in an Incucyte Zoom Live-Cell Analysis System (Essen Bioscience).

**Attachment assays**. 5000 pmFRCs were seeded per well in a 16 chamber E-plate and loaded into a xCELLigence system (both ACEA) maintained in a 37 °C incubator with 5% $CO_2$. Electrical impedance was measured every 10 min until 3 h after seeding.

Kras$^{G12D/G12D}$; p53$^{null}$ MEFs were seeded at 3000 cells per well in a 16 chamber E-plate (ACEA), treated with DIDS and/or EIPA or AHCL and immediately loaded into the xCELLigence system (ACEA) maintained in a 37 °C incubator with 5% $CO_2$. Electrical impedance was measured every 15 min over 4 h post-seeding.

**Statistical analysis**. All viability, proliferation, and attachment assays were assessed using the two tailed unpaired *t*-test with *$P < 0.05$, **$P < 0.01$, ***$P < 0.001$.

**Quantitative reverse-transcription PCR (qRT-PCR)**. RNA extraction was performed using the RNeasy Mini Kit (QIAGEN). One microgram of total RNA was reverse transcribed using the First Strand cDNA synthesis Kit (Thermo Scientific) with oligo(dT) primers. qRT-PCR was performed using TaqMan assays (*ATP2A3* (Mm00443911_m1), *FXYD5* (Mm00435435_m1), *SLC9A1* (Mm00444270_m1)) and a StepOne Real Time PCR System instrument (both Life Technologies).

**Reagents**. 4,4′-Diisothiocyanatostilbene-2,2′-disulfonic acid disodium salt hydrate (DIDS), 5-(N-Ethyl-N-isopropyl)amiloride (EIPA) and Amiloride hydrochloride (AHCL) were purchased from Sigma (D3514, A3085 and A0370000) and dissolved in DMSO.

**Chloride channel blocker analysis**. Kras$^{G12D/G12D}$; p53$^{null}$ MEFs were seeded at 3000 cells per well in black, clear-bottomed 96-well plates and allowed to adhere overnight. Cells were then pre-incubated for 1 h with 5 mM N-(Ethoxycarbonylmethyl)-6-Methoxyquinolinium Bromide (MQAE, Sigma, 46123) before PBS wash and addition of a range of DIDS concentrations in culture media, following which fluorescence intensities (Ex: 350 nm Em: 460 nm) measured by microplate reader (Tecan) over 4 h. Data was normalised to non-MQAE DIDS-treated control.

**Data availability**. All models generated in this manuscript are available for download https://doi.org/10.5281/zenodo.1257326, and are additionally supplied as

Supplementary Data 6. Models can be run using the BioModelAnalyzer (http://biomodelanalyzer.org).

**Code availability**. Code for replicating XGboost machine learning on publicly available gene expression data is available for download https://doi.org/10.5281/zenodo.1257326.

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

## Acknowledgements

We thank the Hall and Shields groups, Alex Gaunt, Stephen Woodhouse, and Phil Jones for useful discussions. Supported by Medical Research Council core funding (D.S., A.R., J.S., E.K., C.P.M.) and the Royal Society (UF130039 to B.A.H.).

## Author contributions

D.S. and B.A.H. conceived the study and wrote the manuscript. D.S. constructed the models and specifications, and performed all analysis. A.R., E.K., L.P., J.S. and C.P.M. performed experimental validations of channel behaviours and analysed resultant data. D.B. and S.S. curated TCGA data and supported the Machine Learning methods. All authors were responsible for the editing of the manuscript.

## Additional information

**Competing interests:** The authors declare no competing interests.

