## [Peer Review File · Nature Communications]

Reviewers' comments:

Reviewer #1 (Remarks to the Author):

In their manuscript on „Stromal osmoregulation controls cellular transformation in cancer and disease“, Hall and colleagues investigate the dependency of osmotic regulation, cell morphology and cancer genesis. They create a qualitative network based on literature search and experimental data. They specifically include and internal pressure sensor, ions and ion transporters as well as disease characteristics. They use reported responses of primary mammalian cells to osmotic changes to specify the expected outcome of the model. Model simulations are performed with BioModelAnalyzer, allowing for qualitative changes with node values being non-negative integers. The model is used to predict the effect of tumor factors in different cell lines as well as the effect of channel specific inhibitors.

The problem posed by the authors is very interesting. Morphological changes, osmotic regulation and cell fate are closely related and deserve to be analyzed in depth. The authors chose an interesting approach that tries to make use of the available information. However, to my opinion, this qualitative model doesn't have the power to explain the intricate regulation between ion fluxes, osmotic pressure and expression of genes.

Many other studies have shown that there is a very sensible interdependency between osmotic pressure, ion and water fluxes, membrane potential and cell size. For example, if osmotic pressure increases and leads to increase in cell size, this will immediately counteract the increase of the osmotic pressure. Ion fluxes lead to change in membrane potential, which in turn regulates the ion fluxes. Any straight arrow indicating just "increase" or "decrease" is therefore much too simple.

Major points.

A quantitative description of the process under study would be extremely important. What prevents from doing this?

Gene regulation is typically only a long-term response to osmotic changes. Therefore, it is only a weak indicator for osmoregulation. How are protein modifications taken into account?

What are the actual numbers of ions transported?

What is the effect of these ion fluxes on the membrane potential?

What are measures for the quality of the model predictions, e.g. presented in Figure 4?

Minor

The legends of most figures do not contain enough information. For example S1 B shows an array in red and green with unreadable entries and the legend states "stability proof progression...".

What do I actually see?

Many figures use some icons in vicinity of proteins. What is the information conveyed by these icons?

Figure 2B shows some simulation time courses. What does an initial state of 0 mean? When do stimuli occur?

Page 13, second paragraph: What are "normal values"?

Figure 4A is supposed to show time courses. Where do I see time in this plot? Are the displayed items experimental data or simulations?

Reviewer #2 (Remarks to the Author):

Shorthouse, Hall and colleagues have developed a new model with the aim of predicting disease states and cell behaviour from input transcriptomic data regarding ion channel and osmotic

dysregulation. The model aims to simplify complex expression data into nodes capable of predicting cell behaviour. The system was validated experimentally using previously published datasets and loss-of-function studies in vitro.

While the model seems very interesting, I do not think the authors have proven their case for stromal osmoregulation controlling cellular transformation in cancer and disease, as per the title. The model seems very interesting, if it can be validated to a high standard, though I must disclose that I am not qualified to assess its construction. The aims were laudable and have the potential to be high impact. Regarding experimental validation, I wish I could be more positive at this time. I found it a challenging paper to review.

Main issues:

1. Lymph node fibroblasts (FRCs) are not a well-supported choice to validate the model. There is no direct evidence that FRCs are under osmotic pressure in tumor draining lymph nodes (TDLN) or LPS treated mice, nor that these changes to a minority of ion transporter/osmotic regulator genes affects FRC phenotype in vivo. There is probably an interesting story here, if carefully explored, but it is currently too undeveloped.

It does not help that only a small minority of genes relevant to osmotic pressure are differently expressed (9/244 osmotic regulatory genes from TDLN FRCs compared to control, and 10/961 probes in LPS treated FRCs). No pathway evidence is cited describing osmotic dysregulation in any model. Expression changes were not validated at protein level despite some careless language ("Transport proteins deregulated.... Proteins are upregulated, downregulated or unchanged" – p18).

Previously published work from these authors (Riedel et al. 2016) shows altered conduit function created by FRCs exposed to tumor-derived factors, but there is no direct evidence that conduit formation by FRCs is altered by osmotic pressure, or that osmotic pressure is a common feature of disease within lymph nodes. The changes in the conduit function were shown to be specific to larger protein transport, not ions.

My overall feeling is that there are probably easier, more established cell systems for validating predictions raised by this model, where osmotic dysregulation is clearly linked to a particular disease and observable cell phenotypes in vivo. The clear link does not yet exist for FRCs. It would be high impact if proven, however.

2. In vitro validation results were difficult to interpret, in their current form.

a. The cells are not exposed to osmotic challenge, despite heavy use of the impact of ion transport and response to hypo/hypertonicity as a rationale in the manuscript, see: the first page of introduction, the construction of the model outlined in Table 1, the first line of results and discussion.

b. Fig 4: Increased attachment is reported after knockdown of Fxyd5, but Supp Fig 4A seems to show that Fxyd5 is not knocked down compared to the Rluc control. A decrease in survival is shown compared to a different (eGFP) control, which has increased expression compared to 2 other controls in Supp 4A. That is, Fxyd5 expression appears increased in the eGFP control, and unchanged in the Fxyd5 knockdown.

c. Slc9a1 and Fxyd5 knockdown each reduce cell survival in vitro (not a reported feature in vivo in TDLN). The authors explain that the Slc9a1 was predicted to be pro-proliferative, and proliferation and viability are linked, which is why knockdown reduces viability. This is not the strongest argument. Proliferation is easily tested in vitro. I would argue that this result does not yet support the model's prediction.

d. In Fig4, the authors report on 2 genes in TDLN, ATP2A3 and FXYD5. The model predicts attachment in vivo, and both genes increase in expression at the late timepoint. But the model predicts that ATP2A3 knockdown will decrease attachment of FRCs in vitro, while FXYD5 knockdown will increase attachment in vitro. This is validated in vitro. But we still do not know the net effect in vivo. I would argue that it is not yet a validation of the model, because the in vitro data suggests there may be pro- and anti-attachment effects operating at once.

e. The model also predicts increased attachment at both timepoints, but ATP2A3 and FXYD5 were both unchanged at the early timepoint. Of the 3 or so genes increased at the early timepoint, which does the model suggest drives increased attachment early, and is this experimentally supported?

3. I have concerns regarding the analysis of microarray data. Very little information is provided on array analyses, some of which are novel (the LPS analysis differs from published work). Please provide full information on analyses.

a. All hand-curated gene lists referred to in-text should be provided in supplementary data.

b. It does not appear that a coefficient of variance (CV) filter has been applied to the tumor-draining dataset, resulting in data being fed to the model based on probes that should be excluded (eg FAM115C, CV for the ctrl sample is 0.76 by my calculation, should be <0.5 at a maximum).

c. It is difficult to understand the criteria for inclusion/exclusion of data. For example, Atp2a3 is an important gene to the paper (profiled in figure 4C). There are 3 probes in the dataset but only one probe is presented in Fig 4A (it appears to be ILMN_2900462). This probe shows no change at the early timepoint and an increase at the late timepoint. Of the other two probes that exist in the dataset, one (ILMN_2638017) had low expression which may be a justification for exclusion (please give details), but the other (ILMN_2688236) shows the opposite - a decrease at D4, no change at D11, with CV<0.5 and high expression. What was the basis for choosing one over the other, and if there is uncertainty with this gene, how does it affect the modelling? Similarly, OPRS1 has 4 probes. 3 of 4 appear to have no significance at P<0.05 at any timepoint, but the 4th is depicted in Fig 4A, and presumably fed to the model. How was this decision made?

d. Appropriate CV filters should be applied to all data prior to analysis. Riedel et al reported in-text that this calculation was performed, but no probes failed the test. This is very improbable for an in vivo dataset and I am reasonably confident it is incorrect based on a glance at the raw data. I hope I am wrong of course, but before this work is published, please ensure the CV calculation used was appropriate to the data - eg the [standard deviation/mean] CV calculation is not appropriate to use for log2 transformed data. Only 9 genes were used as input data for the tumor-draining modelling, so I would imagine that their reliability would be of paramount importance to the modeling.

4. Very important references seem to be misquoted, or at very least the authors' meaning was unclear to me.

a. The authors state that the experimental data from Astarita et al. Nat Imm 2015 supports the model output, but I could not find that this paper examined cell size, contractility, attachment, or migration/membrane dynamics after LPS treatment. It only looks at FRC number.

b. Page 28 states "FRCs exposed to LPS exhibit a reduction in contractility with increased proliferation", citing 2 references that do not seem to support this link, together with a figure reference that does not seem to support the link.

c. Table 2, which cites literature critically important to the modelling, on expected cell behaviour used to validate the model, cites Fletcher et al. 2015 as stating that LPS drives decreased

attachment of FRCs, and that LPS treatment results in no expected change in FRC cell size. I could not find this.

5. It struck me to read that “our model is able to explain the morphological changes in FRC cells exposed to TDLN, solely by taking into account expression changes of ion channels” (p. 13).

How does the model account for existing mechanisms known to drive key FRC phenotypes examined in the paper? Eg, Riedel et al highlighted changes to expression of gap junction proteins as drivers of attachment changes in FRCs. Here this does not seem to be considered.

Similarly, robust evidence now shows that PDPN signalling, subsequent to leukocyte influx and expansion, alters FRC contractility, proliferation, and membrane dynamics (Acton et al. Nature, 2014; Astarita et al. Nat Imm, 2015).

If osmotic regulation contributes or even counters previously reported and hypothesised mechanisms of FRC structural change (eg through stretch-activated ion-permeant channels, such as TRPM7) that would be of interest, but this is yet to be proven and does not appear to be a feature of the modelling. The model appears to treat ion channels as existing in isolation, when in vivo, any effect of ion channels on attachment will occur amongst a cacophony of other molecules driving expression, conformational changes and signalling events for integrins, selectins, cadherins, syndecans, etc.

I regret that my report could not be more positive at this time, as it is a study with a valuable, timely goal and an interesting approach.

Reviewer #3 (Remarks to the Author):

The manuscript entitled 'Stromal osmoregulation controls cellular transformation in cancer and disease' by Shorthouse et al. provides an interesting systemic approach to the role of membrane transporters on tumor progression.

I appreciated the conceptual strategy underlying the osmoregulatory network model: it is globally validated by experimental data from different sources and based on solid literature.

Nonetheless, I have a number of concerns to be addressed in order to consider this paper suitable for publication on Nature Communications.

Major

1. The computational approach is formally suitable and correct, but it needs to be described in more detail. An example: are the variables mentioned at pag 27, line 715 vectorial or scalar, continuous or discrete?

2. I understand the use of MEF as an experimental in vitro model, but the authors should be able to reproduce at least some evidences on human tumor-derived cells from solid tumors (i.e. breast, colon, prostate, renal carcinomas), ideally at different stages of tumor progression.

3. Statistics section for experimental data is completely missing. Please describe in detail the statistical methods, including the tests used to imply significant differences (parametric/non parametric tests...).

Minor

'chlorine' should be 'chloride' in the abstract and introduction.

Reviewer #1 (Remarks to the Author):

In their manuscript on „Stromal osmoregulation controls cellular transformation in cancer and disease“, Hall and colleagues investigate the dependency of osmotic regulation, cell morphology and cancer genesis. They create a qualitative network based on literature search and experimental data. They include specifically include and internal pressure sensor, ions and ion transporters as well as disease characteristics. They use reported responses of primary mammalian cells to osmotic changes to specify the expected outcome of the model. Model simulations are performed with BioModelAnalyzer, allowing for qualitative changes with node values being non-negative integers. The model is used to predict the effect of tumor factors in different cell lines as well as the effect of channel specific inhibitors.

The problem posed by the authors is very interesting. Morphological changes, osmotic regulation and cell fate are closely related and deserve to be analyzed in depth. The authors chose an interesting approach that tries to make use of the available information. However, to my opinion, this qualitative model doesn't have the power to explain the intricate regulation between ion fluxes, osmotic pressure and expression of genes.

Many other studies have shown that there is a very sensible interdependency between osmotic pressure, ion and water fluxes, membrane potential and cell size. For example, if osmotic pressure increases and leads to increase in cell size, this will immediately counteract the increase of the osmotic pressure. Ion fluxes lead to change in membrane potential, which in turn regulates the ion fluxes. Any straight arrow indicating just “increase” or “decrease” is therefore much too simple.

We thank the reviewer for their comments. The aim of this modelling approach, and the wider body of modelling in the logical and executable modelling literature has been to develop useful and usable high level abstractions that are capable of addressing complex systems with limited experimental data, and the roles of rare events. “Qualitative” has numerous definitions within the scientific literature, ranging from discrete systems such as the one described here, to more research in humanities. We chose to use a discrete description of the system, as this formal, mathematical approach allows for quantitative prediction; the discrete levels correspond to ranges of concentrations and activities, and for a given system or assay these thresholds could be quantified. The experimental predictions made in the manuscript reflect the value of the modelling approach despite the necessary approximations. The advantages of this abstraction come from the independence of the higher level function from the specific molecular details, which can allow missing components in a pathway to be omitted from the model. Furthermore, we lack the dependence on inaccessible physical parameters, which are extremely hard to measure for a system this size, and open up the ability to use proof based approaches, which guarantee specific properties of the model. We acknowledge that there are limits to the predictive power of this method. For example, specific concentrations of any molecule in the model are unable to be assessed; we would however expect that false negatives were more common than false positives as transition between states corresponds to a potentially substantial change in concentration. We have extended the section in the introduction (page 4, lines 91 to 99) and added text to the supplementary information (page 1) justifying and explaining the modelling paradigm used.

Major points.

A quantitative description of the process under study would be extremely important. What prevents from doing this?

Quantitative modelling of complex networks such as the one described here would require a substantially more detailed picture of the underlying physical chemistry of the components (i.e. not just concentrations of molecular species, but also kinetic parameters describing the transitions between states). This is not available, as the precise relationships between ions, transporters, and transcriptional effectors in vivo are not known, and in some cases there may be missing links for example between ions and transcriptional regulation where it is not known precisely how they impact expression. As noted above, this is an advantage of the choice of modelling approach used. We further wish to reiterate that this choice of abstraction makes certain analyses available (i.e. model checking) that are not universally possible for quantitative models. As such whilst we agree that a continuous model may have certain benefits our approach has distinct advantages that make it appropriate and valuable. Notes to this effect have been added to the introduction (page 4, lines 91 to 99) and supplementary information (page 1).

Gene regulation is typically only a long-term response to osmotic changes. Therefore, it is only a weak indicator for osmoregulation. How are protein modifications taken into account?

Protein activity in the models is described functionally, and it can for different proteins encapsulate expression, phosphorylation status, conformation/binding partners, lipid interactions etc. that drive the underlying activity. As such the specific predictions that are made must take account of this explicitly; an example of this is the regulation of sodium-potassium ATPase by cytoskeletal interactions, protein subunits, and small molecules. Here the expression and phosphorylation of various binding partners, as well as its own expression level determines its activity, in particular, the protein is separated into two domains within the model, a domain that involves transportation of ions, and a domain that involves interactions with the cytoskeleton, as they are known to be functionally distinct. We have added notes describing this phenomenon, including sodium-potassium ATPase as a specific example (page 15, lines 383 to 387, and Figure 4, B). We further note that we can (and in previous studies, have) explicitly represent different modifications states with distinct nodes (un-phosphorylated and phosphorylated protein for example) but it has not been necessary here.

What are the actual numbers of ions transported?

What is the effect of these ion fluxes on the membrane potential?

As described above we, our model represents global (within a single cell) activity and concentration levels, and as such we do not, and cannot calculate explicit numbers of ions, or exact membrane potential. The strength of the technique chosen for this study is that it can handle “incompleteness”, but still give useable answers (“This protein significantly impacts X behavior”), and correctly generate emergent properties such as cellular phenotype, but at the cost of exact measurable concentrations.

What are measures for the quality of the model predictions, e.g. presented in Figure 4?

We define a specification that states the model must be stable, and that individual variables in the stable state have specific assignments. By “stable” we mean that the model has a global, fix-point attractor. In this case each cellular phenotype (proliferation, migration, adhesion, contractility, cell size) is found to stabilize at a particular value under each condition (ETDLN, LTDLN, LPS, HOM-MEF). We compare these phenotypes to the specification generated from the literature to assess the quality of the model. Because the model results in a discrete answer for whether a phenotype is increased, decreased, or unchanged, we chose to represent it as a heatmap. We compare the model output to a specification generated from the literature, pathway analysis, or through previous experiments. Note that whilst experiments may predict that proliferation for example doubles, the network will simply predict that it is increased. We have clarified this in the text (page 10/11, lines 273 to 291), and added to the supplementary information, (page 1).

Minor

The legends of most figures do not contain enough information. For example S1 B shows an array in red and green with unreadable entries and the legend states “stability proof progression...”. What do I actually see?

We thank the reviewer for highlighting this issue; we have expanded the figure legends throughout the text. Specifically for the stability proof progression; rather than explicitly simulating the system in all states we use a proof algorithm presented in Cook et al (2010) implemented in the BMA. In this algorithm the proof of stability is achieved by reasoning over the behavior of individual variables repeatedly until either all variable ranges can be reduced to a single value or all possible approaches are exhausted. This is the information that is presented in the proof progression; the series of variable range reductions until the global fix point is reached. A SAT solver is used to symbolically prove the stability or find counter examples if stability has not been proved, using the reduced ranges calculated in the first step. We have added notes describing how the algorithm works to the supplementary information, and simplified the figure (Now figure S3)

Many figures use some icons in vicinity of proteins. What is the information conveyed by these icons?

We apologise for the lack of description here – the icons represent the 3 dimensional structure of the protein generated by each gene, and are added only for illustrative purposes, as the icons occur in the schematic of the model, and in the pathway schematics. We have changed the tables in figures 5 and 6 to better explain this.

Figure 2B shows some simulation time courses. What does an initial state of 0 mean? When do stimuli occur?

We have updated the figure (figure 2, B) to more clearly show when stimuli were activated. In all presented simulations, all variables start from an assignment of 0 (unless explicitly changed), the model then “normalizes”, whereby nodes adjust to their steady-state values. Simulation time course plot starts from this point. At this point a stimulus is added, and the steady-state is perturbed. We also note that this is a distinct analysis from the stability proof described above, as this shows temporal ordering of

events. We have added details to the text to further explain this (page 12, lines 302 to 307).

Page 13, second paragraph: What are “normal values”?

We apologize for the omission – in this case normal values represents “normal conditions”, conditions in which concentrations of external ions are unchanged from the homeostatic norm. In this state the cell feels no great osmotic pressure from the increased presence or absence of external osmolytes. We have altered the text to clarify this (page 17, lines 419 to 420).

Figure 4A is supposed to show time courses. Where do I see time in this plot? Are the displayed items experimental data or simulations?

Figure 4A represents gene expression levels at 3 different times, 0 hours after exposure to tumor factors (a control), 4 hours after exposure (Early Tumor Draining Lymph Node cells – ETDLN), and 11 hours after exposure (Late Tumor Draining Lymph Node cells – LTDLN). Whilst this represents snapshots of the expression of genes in the same cells at different time points after exposure to tumor factors, this is not true timecourse data, we have edited the text to clarify this.

Reviewer #2 (Remarks to the Author):

Shorthouse, Hall and colleagues have developed a new model with the aim of predicting disease states and cell behaviour from input transcriptomic data regarding ion channel and osmotic dysregulation. The model aims to simplify complex expression data into nodes capable of predicting cell behaviour. The system was validated experimentally using previously published datasets and loss-of-function studies in vitro.

While the model seems very interesting, I do not think the authors have proven their case for stromal osmoregulation controlling cellular transformation in cancer and disease, as per the title. The model seems very interesting, if it can be validated to a high standard, though I must disclose that I am not qualified to assess its construction. The aims were laudable and have the potential to be high impact. Regarding experimental validation, I wish I could be more positive at this time. I found it a challenging paper to review.

The aim of this paper was to establish a formal model of the relationships between ion channels, water flow and cellular behavior, and to validate aspects of the model by reference to two experimental systems. Whilst we concede that further experiments could be performed to validate different aspects of the model, it is beyond the scope of this paper to explore all aspects. To better address this issue we have nevertheless

- Modified the title, abstract, and text to better reflect the results in the paper
- Used a machine-learning driven analysis of available cancer datasets to further demonstrate the role of channel regulation in cancer
- Extended the experimental validation presented in the paper

These modifications are detailed in specific responses to the reviewer below

Main issues:

1. Lymph node fibroblasts (FRCs) are not a well-supported choice to validate the model. There is no direct evidence that FRCs are under osmotic pressure in tumor draining lymph nodes (TDLN) or LPS treated mice, nor that these changes to a minority of ion transporter/osmotic regulator genes affects FRC phenotype in vivo. There is probably an interesting story here, if carefully explored, but it is currently too undeveloped.

It does not help that only a small minority of genes relevant to osmotic pressure are differently expressed (9/244 osmotic regulatory genes from TDLN FRCs compared to control, and 10/961 probes in LPS treated FRCs). No pathway evidence is cited describing osmotic dysregulation in any model. Expression changes were not validated at protein level despite some careless language (“Transport proteins deregulated.... Proteins are upregulated, downregulated or unchanged” – p18).

We understand the reviewers concerns regarding FRCs as a validation of the model, however we are not assuming that the systems in question (tumor draining lymph nodes, and LPS exposed lymph nodes) are exposed to osmotic stress. All simulations and analyses presented (except those explicitly validation osmotic network homeostasis – Figure 2) are performed assuming no osmotic stress on the cell. We do not claim to have proven that osmotic stress is a uniquely transformative agent in FRCs, but claim that disruptions to the osmotic regulation network, and the broader associated ion transport pathways, can partly explain and predict the cellular transformations observed. This is because many cellular properties (attachment, membrane stability, cell cycle checkpoint gating etc.) are performed and controlled through the activity of ion channels, though we do not claim they are the sole transformative control – they are likely controlled by transcription factors and other signaling networks. We focused initially on the osmotic regulation network due to its robustness, ubiquity, and self-containment compared to other networks (calcium signaling, pH maintenance etc). An important distinction is that we do not claim to have elucidated an exact mechanism, or higher level transcriptional control, we rather show that ion channels can act as the “agents of change”, and that by studying their activity, we can predict cellular properties, using lymph node fibroblasts as a validation system.

Whilst we agree that 9/244 and 10/961 significantly dysregulated genes seems low, when applied to the osmotic regulatory network in Figure 2, it results in a disruption to 4/8 of the major classes of primary osmotic response proteins, the fact that this disruption does not lead to significant long term changes in cell size (something supported by the model) attests to the robustness of the network. From this we feel that the osmotic regulatory network was a good starting point for explaining behaviour – especially given that proteins in the network (slc9a1 for example) are well understood to have implications on cellular properties beyond regulating osmotic stress. We have changed the title and made it more clear in the text that we are not specifically claiming to elucidate osmotic stress as the driver of transformation.

We note the reviewers concern regarding the use of expression data, and we thank them for highlighting the issue. As noted in responses to the previous reviewer, the control of channel and transporter activity is not limited to the expression levels of the protein; the sodium potassium ATP-ase for example is also controlled by FXYD proteins, as well as cytoskeletal interactions, among other factors. Channel opening/closing may be further controlled by binding partners, post translational modification, and subcellular localization and as such we would not expect this reflected necessarily in the expression data alone.

We have altered the title to better reflect the drive of the study, and edited the text in numerous places (For example page 3, lines 76-78, page 12, lines 309-312)

Previously published work from these authors (Riedel et al. 2016) shows altered conduit function created by FRCs exposed to tumor-derived factors, but there is no direct evidence that conduit formation by FRCs is altered by osmotic pressure, or that osmotic pressure is a common feature of disease within lymph nodes. The changes in the conduit function were shown to be specific to

larger protein transport, not ions.

We do not claim to have elucidated osmotic pressure as a driver of transcription change – we have:

- Recognized that some of the dysregulated genes observed within lymph node transformation correspond to ion channels – in particular involved in the osmotic regulation network
- Constructed a network of osmotic regulation that is robust in the ways suggested through experiments
- Expanded the network to include core ion transport domains and their impacts on cellular phenotypes
- Disrupted the network in the same way observed in varying conditions (TDLN, LPS exposed lymph node, MEF loss of homozygosity in *Kras*).
- Shown that the model accurately predicts the cellular phenotype, and has predictive power to control cell behavior

The specific dysregulation of ion channel genes observed within the network is not consistent with cells being exposed to osmotic stress, and is likely caused by exposure to more complex signaling molecules or metabolites and is currently not understood.

My overall feeling is that there are probably easier, more established cell systems for validating predictions raised by this model, where osmotic dysregulation is clearly linked to a particular disease and observable cell phenotypes in vivo. The clear link does not yet exist for FRCs. It would be high impact if proven, however.

We thank the reviewer for this comment. We have performed machine learning on publically available datasets and highlight that ion channels are a possible distinguisher of different cancer types, and that osmotic machinery is one of the top features used by the algorithm to distinguish between a cancer and normal cell, through expression alone. We have also studied and validated a drug treatment on cells from a completely separate cell line (MEFS). We expect that many systems can be understood in this way, as any system where a cell changes attachment ability, migrates differently, proliferates more etc. can be understood in terms of ion transport. All these cellular processes involve ions as an essential component (attachment relies on calcium and pH to control actin and other fillaments, proliferation relies on concentrations of chlorine and sodium at specific points of the cell cycle, migration is caused by influx of ions and water at a leading edge and expulsion from a trailing edge etc.), and characterizing the change in ion fluxes and concentrations allows estimates of how these behaviors differ, and predictions of how to disrupt them. The osmotic regulation network acts as a “hub” where multiple ion types are involved in a single process, and as such is a place to begin building a network of ion transport from. The fact that early and late stage tumor draining conditions lead to 3 out of the 8 core osmotic regulation protein types to be differently expressed shows that it is an appropriate network to begin with in terms of studying ion transport within these cells.

We accept that other systems may involve osmotic stress directly impacting cell phenotype, rather than simply differential expression of genes in the network, and it would be of interest to study these in the future.

2. In vitro validation results were difficult to interpret, in their current form.

a. The cells are not exposed to osmotic challenge, despite heavy use of the impact of ion transport and response to hypo/hypertonicity as a rationale in the manuscript, see: the first page of introduction, the construction of the model outlined in Table 1, the first line of results and discussion.

We do not believe that FRCs are being exposed to significant osmotic stress in lymph node. Whilst the response of the osmotic regulation network to stress is calculated and tested, this was performed to ensure that the network responds similarly to the literature, not as a proof that this response is occurring in the systems we go on to study. Validating the physiological behavior of the model is an essential component of model building, and due to centrality of the osmotic regulatory network to the greater model, it is imperative that we prove the network functionally recapitulates biological homeostasis. We did not expose cells to osmotic challenge because we do not believe that they are experiencing osmotic challenge *in vivo*, simply that the osmotic regulatory network of proteins is dysregulated, and that this dysregulation can influence other cellular behaviors such as attachment and contractile ability.

b. Fig 4: Increased attachment is reported after knockdown of Fxyd5, but Supp Fig 4A seems to show that Fxyd5 is not knocked down compared to the Rluc control. A decrease in survival is shown compared to a different (eGFP) control, which has increased expression compared to 2 other controls in Supp 4A. That is, Fxyd5 expression appears increased in the eGFP control, and unchanged in the Fxyd5 knockdown.

We thank the reviewer for this observation, we have re-done this analysis after confirming knockdown of *Fxyd5* against both controls (Figure S6, A).

c. Slc9a1 and Fxyd5 knockdown each reduce cell survival in vitro (not a reported feature in vivo in TLDN). The authors explain that the Slc9a1 was predicted to be pro-proliferative, and proliferation and viability are linked, which is why knockdown reduces viability. This is not the strongest argument. Proliferation is easily tested in vitro. I would argue that this result does not yet support the model's prediction.

We thank the reviewer for this comment, they are correct to point out the discrepancy in proliferation vs viability. Historically in the primary literature the terms proliferation and viability have been used interchangeably. Often simple counts of live cells are used to determine the measure of proliferation occurring in culture, when a decrease in cell count could potentially imply a greater apoptotic

occurrence rather than decreased proliferative capacity. Additionally decreases in viability don't necessitate a decrease in proliferation rate. In the literature, often proliferation is used in place of viability. Given that the model was based on literature, we took specific terms from the literature to generate the model. Due to the inconsistencies observed, prompted by the comment from reviewer #2, we believe that the behavior most likely encapsulated in the model is viability. Viability is a more general term, and includes a measure of both proliferative capacity, and also death rate. Nevertheless we have performed proliferation assays to measure proliferation of cells after knockdown of *slc9a1* and *fxyd5*. We find that proliferation (measured by live cell count) of FRCs is reduced after knockdown of *fxyd5*, indicating that the correspondent viability loss is partly caused by a reduction in proliferation rate. However, knockdown of *slc9a1* did not result in any change in proliferation rate, but a significant decrease in viability, this is supported in the literature, and suggests it is an enabler of proliferative ability, or protector against apoptosis, but not a driver of cellular proliferation.

It is an inherent feature of the technique (and computational modelling techniques in general) that they are reliant on reports in the literature to construct and validate models. In this case, proliferation and viability had been intertwined and often reported with the same measurements, and upon careful assessment, we think our model best represents viability, due to the less specific phenotypes encapsulated in this property. We have changed the wording of the manuscript to reflect this change, included proliferation assays in the manuscript, and added a section to the supplementary information discussing why viability was chosen over proliferation.

d. In Fig4, the authors report on 2 genes in TDLN, ATP2A3 and FXYD5. The model predicts attachment in vivo, and both genes increase in expression at the late timepoint. But the model predicts that ATP2A3 knockdown will decrease attachment of FRCs in vitro, while FXYD5 knockdown will increase attachment in vitro. This is validated in vitro. But we still do not know the net effect in vivo. I would argue that it is not yet a validation of the model, because the in vitro data suggests there may be pro- and anti-attachment effects operating at once.

Whilst it is correct that there are pro and anti-attachment proteins acting simultaneously, the model predicts that a wide range of factors that influence attachment properties. In particular, cellular potassium levels can influence attachment, and disruption of intracellular calcium homeostasis through other mechanisms can, and in the model are predicted to influence attachment behavior. The model predicts that the overall outcome is an increased, based upon weighting each influence equally (3 increasing contributors, and 1 decreasing).

e. The model also predicts increased attachment at both timepoints, but ATP2A3 and FXYD5 were both unchanged at the early timepoint. Of the 3 or so genes increased at the early timepoint, which does the model suggest drives increased attachment early, and is this experimentally supported?

Of the 5 channel associated genes altered during the early stage tumor exposed FRCs, two are predicted to have a pro-attachment influence, *fam115c* due to its interactions with the TRP family of calcium transporters, and *slc9a1*, a sodium/proton symporter. *Slc9a1* in particular is well supported as driving attachment changes through pH alteration of the extracellular matrix and glycocalyx (Putney 2002, Stock 2006), and is also shown to reduce attachment in MEFs when it is targeted with inhibitors in the later portion of the manuscript (Figure 6).

3. I have concerns regarding the analysis of microarray data. Very little information is provided on array analyses, some of which are novel (the LPS analysis differs from published work). Please provide full information on analyses.

We have included most details of the microarray analysis. We also note the microarrays are used as a guide, and suggest targets that we more thoroughly studied through literature review, before including them in the model. We are aiming to provide a qualitative overview of the activity of channels and their impact on cellular behavior, validated through experimental manipulation.

a. All hand-curated gene lists referred to in-text should be provided in supplementary data.

We apologise for this omission, and have included all gene lists in the supplementary.

b. It does not appear that a coefficient of variance (CV) filter has been applied to the tumor-draining dataset, resulting in data being fed to the model based on probes that should be excluded (eg FAM115C, CV for the ctrl sample is 0.76 by my calculation, should be <0.5 at a maximum).

Data normalization was performed through standard methods in the limma package for R. In particular technical artefacts were corrected for, background correction was applied before statistical analysis was performed. Whilst a low coefficient of variance is a useful indicator that a probe is potentially reliable, it is not necessarily an exclusionary filter. We have nevertheless tested the model with FAM115C unchanged under both early and late stage conditions and note that all phenotypes remain, indicating that upregulation of this gene alone is not sufficient to alter phenotype.

*c. It is difficult to understand the criteria for inclusion/exclusion of data. For example, *Atp2a3* is an important gene to the paper (profiled in figure 4C). There are 3 probes in the dataset but only one probe is presented in Fig 4A (it appears to be ILMN_2900462). This probe shows no change at the early timepoint and an increase at the late timepoint. Of the other two probes that exist in the dataset, one (ILMN_2638017) had low expression which may be a justification for exclusion (please give details), but the other (ILMN_2688236) shows the opposite - a decrease at D4, no change at D11, with CV<0.5 and high expression. What was the basis for choosing one over the other, and if there is uncertainty with this gene, how does it affect the modelling? Similarly,*

OPRS1 has 4 probes. 3 of 4 appear to have no significance at $P < 0.05$ at any timepoint, but the 4th is depicted in Fig 4A, and presumably fed to the model. How was this decision made?

We thank the reviewer for this comment – traditionally the different probes that map to the same gene within an illumina microarray map to different locations within the gene model. These may correspond to exons that may result in alternative splice variants. In addition, due to annotation errors in databases used when the chips was designed, some illumina probes do not map to the purported location defined by the annotation (Reference our reannotation of illumina arrays paper). Generally a single probe is enough to imply that the transcription of a gene is increased. In the case of *Atp2a3* it has 7 splice variants according to ensembl, so probe differences are likely due to expression of different variants of the protein. Studying the probe at chromosomal level in ensemble does not allow us to distinguish this difference, as the splice variants are fairly similar. The model is a qualitative representation of the state of the cell at a particular point in time, derived from the literature, and we chose to model only the most significantly dysregulated probes in our system. After applying p value cutoffs and a Fold Change cutoff of ± 1.5 we are ensured that only the most significantly altered probes are taken into account. The probe from *Atp2a3* showing a decrease at day 4 does not meet this cutoff criterion and so was excluded from the model. Whilst this cutoff is arbitrary it serves to ensure that only the most impactful and reliably changed genes are considered for analysis.

d. Appropriate CV filters should be applied to all data prior to analysis. Riedel et al reported in-text that this calculation was performed, but no probes failed the test. This is very improbable for an in vivo dataset and I am reasonably confident it is incorrect based on a glance at the raw data. I hope I am wrong of course, but before this work is published, please ensure the CV calculation used was appropriate to the data – eg the [standard deviation/mean] CV calculation is not appropriate to use for log2 transformed data. Only 9 genes were used as input data for the tumor-draining modelling, so I would imagine that their reliability would be of paramount importance to the modeling.

Dispersion analysis and normalization was performed on the array published in Riedel et al using standard methods from Bioconductor (supplementary methods of that paper), specifically Variance Stabilization Transformation, and quantile normalization. Riedel et al report a coefficient of variation between samples measuring an average over all probes (informing on the reliability of the biological repeats overall, rather than specific probes). It is also worth noting that the microarray deals with a complex biological phenomena including timecourse of exposure to tumor factors, and so a greater variability is to be expected compared to microarrays dealing with for example, late stage cancer vs normal cells. Samples from early stage tumor conditions are likely to have more variance purely because of biological differences between tumor exposed lymph nodes, particularly because we expect it to be in a transitional period between two steady states – normal, and a tumor transformed lymph node.

4. Very important references seem to be misquoted, or at very least the authors' meaning was unclear to me.

a. The authors state that the experimental data from Astarita et al. Nat Imm 2015 supports the model output, but I could not find that this paper examined cell size, contractility, attachment, or migration/membrane dynamics after LPS treatment. It only looks at FRC number.

We apologize for the confusion here; we have rewritten this section of the methods to better reflect the findings, including expanding the references used – in particular expanding the justification for the specification, and removing attachment from it entirely, whilst there is some inferred attachment behavior, particularly in light of PDPN expression and activation, there is no specific measure of whether it is increased or decreased within LPS exposed LN.

b. Page 28 states “FRCs exposed to LPS exhibit a reduction in contractility with increased proliferation”, citing 2 references that do not seem to support this link, together with a figure reference that does not seem to support the link.

We apologize for this mistake, we have updated this section of the manuscript.

c. Table 2, which cites literature critically important to the modelling, on expected cell behaviour used to validate the model, cites Fletcher et al. 2015 as stating that LPS drives decreased attachment of FRCs, and that LPS treatment results in no expected change in FRC cell size. I could not find this.

We thank the reviewer for this comment – we have updated this section of the manuscript, including removing attachment and cell size predictions from the specification, due to lack of experimental evidence on their change during LPS exposure.

5. It struck me to read that “our model is able to explain the morphological changes in FRC cells exposed to TDLN, solely by taking into account expression changes of ion channels” (p. 13).

How does the model account for existing mechanisms known to drive key FRC phenotypes examined in the paper? Eg, Riedel et al highlighted changes to expression of gap junction proteins as drivers of attachment changes in FRCs. Here this does not seem to be considered.

Similarly, robust evidence now shows that PDPN signalling, subsequent to leukocyte influx and expansion, alters FRC contractility, proliferation, and membrane dynamics (Acton et al. Nature, 2014; Astarita et al. Nat Imm, 2015).

The reviewer raises an important point here. Firstly, we stress that we do not claim here or in the paper to question any previous claims regarding known pathways that influence cellular behavior, in particular PDPN which has been shown numerous times to influence FRC phenotype. Our work presented in the paper focuses on the mechanistic requirement for ion channel involvement in cellular behaviors such as proliferation, movement, contraction etc. We do not claim that activation of channels are a primary instigator of phenotypic change, but that they are commonly activated in predictable ways in response to unknown, upstream signaling events in cellular decision making. In the case of PDPN for example, it is unknown specifically how PDPN reduces contractility, but it is thought to interact with ERM family

proteins, which are known to also interact with NHERF, a cofactor associated with modulation of numerous membrane proteins, including calcium channels and Na-K-ATPase, which is deregulated in the microarray. Changes in ionic concentrations as a result of these interactions can influence cell behavior such as contraction. We do not explicitly consider this pathway interaction due to lack of specific evidence, but nevertheless the model predicts a reduction in contractility occurs, partly due to the downregulation of subunits of Na-K-ATPase, and resultant effects on sodium activated calcium channels (among other phenomena). Ion channels do not act alone, but in most cases we do not know enough about signaling events leading to their expression and activation to model them in their myriad of pathways (and if we did – the model as it stands is on the boundary of being intractable due to its size and computational intensity, so any increase in complexity will necessarily mean an inability to look at whole cell dynamics). We concede that we are not modelling the entire cascade involved in, for example contractility changes, but we find that modelling ion dependent behaviors is enough to predict the phenotype change correctly, and to find potential intervention sites to change that phenotype without the full understanding of entire network. We have added notes to this effect in the main body of the text (page 4, lines 93-95) and supplementary (page 1).

If osmotic regulation contributes or even counters previously reported and hypothesised mechanisms of FRC structural change (eg through stretch-activated ion-permeant channels, such as TRPM7) that would be of interest, but this is yet to be proven and does not appear to be a feature of the modelling. The model appears to treat ion channels as existing in isolation, when in vivo, any effect of ion channels on attachment will occur amongst a cacophony of other molecules driving expression, conformational changes and signalling events for integrins, selectins, cadherins, syndecans, etc.

Fundamentally, a systems biology approach allows us to attempt to tackle this complexity formally by encoding the interactions and causal relationships between a diverse range of pathways. We have explicitly considered a diverse range of proteins in our model, and this in turn has allowed us to link ion channel coordination to cellular behavior. Whilst we do not doubt that there are other interactions in the cell that can control behavior- and have not stated otherwise- our model and experimental validations show how coordination between ion channels can play a role in that. The importance and wider impact of this should be seen in the context that ~50% of drugs target membrane proteins- it both implies that efficacy of drugs can be determined by multiple activities (in the case of NHE1 inhibitors) and synergies between such drugs are available. We have added notes to this effect to the main body of the text (page 28, lines 705-708).

I regret that my report could not be more positive at this time, as it is a study with a valuable, timely goal and an interesting approach.

Reviewer #3 (Remarks to the Author):

The manuscript entitled 'Stromal osmoregulation controls cellular transformation in cancer and disease' by Shorthouse et al. provides an interesting systemic approach to the role of membrane transporters on tumor progression.

I appreciated the conceptual strategy underlying the osmoregulatory network model: it is globally validated by experimental data from different sources and based on solid literature.

Nonetheless, I have a number of concerns to be addressed in order to consider this paper suitable for publication on Nature Communications.

Major

1. The computational approach is formally suitable and correct, but it needs to be described in more detail. An example: are the variables mentioned at pag 27, line 715 vectorial or scalar, continuous or discrete?

The model is formally defined in the in the supplementary information, but for reference the individual variables are integers with a finite range. We have added a note to this effect in the text (page 4, lines 95 to 96).

2. I understand the use of MEF as an experimental in vitro model, but the authors should be able to reproduce at least some evidences on human tumor-derived cells from solid tumors (i.e. breast, colon, prostate, renal carcinomas), ideally at different stages of tumor progression.

We thank the reviewer for this suggestion. Addressing the role of ion channels as a general property of cancer is non-trivial due to the massive diversity in cancer types, even before stages of cancer progression are considered. However, this question speaks to the generality of the conclusions drawn here and to attempt to answer it we have added additional work on publically available datasets of human tumors TCGA, showing that membrane protein deregulation is common, we additionally show that using gradient boosted decision trees allows classification of single cell samples into cancer and non-cancer subtypes, and show that extracting feature weights from this analysis return the genes that most contribute to this distinction. This is consistent with observations in the literature where many of these have been previously implicated in cancers. Furthermore many are involved in the osmotic response, including NHE and AE channels, which we show reduce proliferation when synergistically knocked down with drugs in MEFs (pages 6 and 7).

3. Statistics section for experimental data is completely missing. Please describe in detail the statistical methods, including the tests used to imply significant differences (parametric/non parametric tests...).

We apologize for this omission. We have added the relevant details, (pages 37 and 38, lines 1013 to 1015, and 1054 to 1056).

Minor

'chlorine' should be 'chloride' in the abstract and introduction.

We have made this correction.

Reviewers' comments:

Reviewer #1 (Remarks to the Author):

I highly appreciate the motivation to create a model of osmotic stress response despite the fact that many quantitative data are not at hand. As such, the choice to create a qualitative model is understandable. However, the question is still how good the predictions are that are made by the model. And here, we come back to ask for quantitative properties.

Let me illustrate this with an example: In figure 2 is presented the responses to hypertonic and hypotonic stimuli. It is known for long time (e.g. many papers by the Posas group, for example de Nadal et al., EMBO reports 2002) that cells shrink under hypertonic stress and then slowly adapt to gain normal volume, while they swell (or even rupture) under hypotonic treatment. Both responses happen immediately (within seconds). However, what figure 2 shows is exactly the opposite: after some simulation time, cells get large under hypertonic stimulus they shrink temporarily. So, here the prediction appears to be simply wrong.

The model also makes "predictions" about a series of processes around proliferation and cell size. Model implementation is described in Table S3, but none of the effects is demonstrated. On the opposite, Table S3 states, among others, that internal Calcium increases attachment. However, Figure 5 demonstrates the opposite.

A drawback of the representation is that there is no way to play around with the model since there is neither an executable version available nor a complete description.

Reviewer #2 (Remarks to the Author):

I'm pleased to say I have come to appreciate this manuscript a lot more than I did originally, provided certain constraints are made very obvious to readers.

My original concern was that the functional validation is weak by cancer biology or immunology standards for a journal of this calibre, with no in vivo verification of the molecule/function relationships described, just broadly aligned with previously published in vivo observations. But I take the authors' point in rebuttal that the validation is intended to validate the model's performance rather than define new mechanistic relationships. The value of the work lies in the model, which is interesting, timely and novel and represents significant progress. However, so much of the paper is devoted to its validation, that it is necessary to make this distinction much clearer, and to readers from a broad range of specialities. I happily admit I misjudged its intended significance originally, but I am confident I would not be alone in that, so it is something to consider.

It is important to carefully avoid mechanistic overstatement, because identifying novel regulators of cellular attachment etc in novel cell types and disease states has not been achieved to a sufficient standard in my opinion (and the authors state in rebuttal that describing mechanism this is not their intent). For example, I would like to see the title of the paper mention the predictive model. Currently it focuses the reader towards molecular function in disease states. Cell lines are not diseases, and the interesting t-SNE work on human cancers is supportive but entirely correlative. The model is the novel aspect of this paper, far more than the effects of ion channel networks on cancer or in cultured FRCs, which is weakly proven. But I appreciate the point that the data should be seen here solely as a system of in vitro validation for the model's predictions. Provided it is clear that these molecular functions are yet to be validated in vivo, I agree that the data fulfils the intended purpose. The writing now needs to better reflect this intention.

The validation is also somewhat weakened because effects of single proteins are mostly viewed in isolation rather than net effects of multiple proteins, which is the model's greatest advantage over literature searches and other traditional single molecule discovery approaches (so its performance in this respect is of great interest). But the work describing successful co-inhibition of SLC4 and SLC9 is an important validation and very interesting. I do still think it should be validated in vivo at least once before we can be confident in the model's performance and future relevance (particularly in light of the title), but I leave that to the discretion of the editorial team.

Minor points:

Line 677: "LPS exposure represents changes associated with the LN immune response, and is an entirely different physiological stimulus to TFs. We find that our model accurately predicted the cellular responses displayed in the literature" – table 3 reports no prediction from literature for the effect of LPS on FRCs.

Line 705: "Whilst we do not claim to completely encapsulate all cellular behaviour, or the only mechanism for its change, we have shown that focussing on ion channels is enough to partly explain cellular morphological change, and provide useful predictions for modulation of cellular behaviour." – mention of in vitro limitations to the work would improve the interpretation here.

Line 668: "Stromal cells are highly integral to this process, ... a mechanism has not previously been elucidated. We propose that the reorganisation of ion channel expression is one of the key mechanisms by which cellular transformation of TDLN FRCs occurs. We have shown that targeted knockdown of certain channels can modulate behaviour in specific ways within FRCs." – clear mention of in vitro limitations to the work would improve the interpretation here.

Line 648: "and show that dysregulation of the membrane transport network can explain and impact the behaviour of cells in disease such as cancer." – "may explain and impact" would be more accurate here.

Reviewer #3 (Remarks to the Author):

The paper is now suitable for publication.

Reviewer #1 (Remarks to the Author):

I highly appreciate the motivation to create a model of osmotic stress response despite the fact that many quantitative data are not at hand. As such, the choice to create a qualitative model is understandable. However, the question is still how good the predictions are that are made by the model. And here, we come back to ask for quantitative properties. Let me illustrate this with an example: In figure 2 is presented the responses to hypertonic and hypotonic stimuli. It is known for long time (e.g. many papers by the Posas group, for example de Nadal et al., EMBO reports 2002) that cells shrink under hypertonic stress and then slowly adapt to gain normal volume, while they swell (or even rupture) under hypotonic treatment. Both responses happen immediately (within seconds). However, what figure 2 shows is exactly the opposite: after some simulation time, cells get large under hypertonic stimulus they shrink temporarily. So, here the prediction appears to be simply wrong.

Firstly, we regret that, during iteration of the manuscript, a mistake was introduced involving the labels on Figure 2. They were incorrectly switched in the process of redrawing as a response to review, we have included the original figure to illustrate that it was correct in the primary version of the manuscript. We have corrected the figure, and made it clear in the text explicitly how the model behaves (lines 292-296). Additionally the models are available at https://github.com/shorthouse-mrc/biomodelanalyzer_ionchannels, and these specific traces can be reproduced by importing each model (hypertonic conditions, and hypotonic conditions) into the BMA at www.biomodelanalyzer.org and running a simulation analysis (the middle button on the right-hand side of the window. Observing the trace for cell size, after an initial stabilization period, will result in the traces in Figure 2. This is an extremely important note as this was not a prediction of the model, but rather used a test of model correctness under several conditions (e.g. change from non-draining to draining fibroblast reticular cells). The technical terminology for this type of test for model validity- a formal specification- was used in the text, and has been edited to make this clearer (lines 241-242).

Secondly, we stress that the choice of a discrete model is not solely motivated by available data. **As noted explicitly in the text** (lines 100-102) the choice of a discrete abstraction makes several analyses possible that are not generally available in continuous systems. As we are able to address the entirety of state space we are able to make universal statements- such as this will **never** happen, or this will **always** happen- that are not generally possible in ODE models. This fundamental feature of the chosen abstraction is a motivating reason even if accurate quantitative parameters were available. Indeed, this has been the motivation behind the field of “logical modelling” for two decades – this is also detailed in the supplementary information.

Thirdly, the choice of abstraction addresses an increasingly recognised issue in mathematical modelling of how to best address complexity in biological systems, recently reviewed by Gosak et al, (<https://doi.org/10.1016/j.plrev.2017.11.003>), and Gómez-Schiavon and Hana El-Samad (<https://doi.org/10.1101/248419>). Small, highly abstract continuous models of behaviour may give insights into general principles of biological behaviour but downplay complex interactions, and have been known to suffer from issues of parameter degeneracy (where an experiment cannot resolve individual parameters but can identify functions of multiple parameters). In contrast, even relatively well parameterised complex network have missing features and may require the use of parameters measured in conditions that do not reflect the conditions or cell line of interest. Our choice of abstraction allows us to include the complex network topology whilst effectively coarse graining the variable update functions.

The model also makes “predictions” about a series of processes around proliferation and cell size. Model implementation is described in Table S3, but none of the effects is demonstrated. On the opposite, Table S3 states, among others, that internal Calcium increases attachment. However, Figure 5 demonstrates the opposite.

We thank the reviewer for highlighting this apparent discrepancy. The supplementary table refers to Ko et al (2001) that discusses cell-cell adhesion, driven by cadherin activation in response to calcium rises. This is in contrast to focal adhesion kinase mediated attachment, discussed in Giannone (2004), where a rise in calcium causes a loss of attachment to surfaces through the disassembly of focal adhesion kinases (FAK). Both effects are included in the model, though attachment as modelled is mediated by primarily by FAK. As such, the validated prediction of the model, that loss of ATP2A3 increases cytoplasmic calcium and thereby reduces attachment is consistent with the model specification. To remove any ambiguity in table S3 we have added an additional column to show how users can modify the model to reproduce the behaviour described.

A drawback of the representation is that there is no way to play around with the model since there is neither an executable version available nor a complete description.

Upon submission of the manuscript, we included the model along with the manuscript for review (“supplementary code”). To further simplify reproducibility, we have separated distinct cell type models and made them available at https://github.com/shorthouse-mrc/biomodelanalyzer_ionchannels in addition to adding them as a zip to the manuscript. We further note that the BioModelAnalyzer is open source under the MIT license (<https://github.com/Microsoft/BioModelAnalyzer>) and the user interface is publically available at www.biomodelanalyzer.org. We have further expanded the detail describing the model in the supplementary information, including details on how cellular phenotypes are controlled in the model (Supplementary Table 3), and justification for gene interactions included in the model (Supplementary Table 5). We note that in the process of reconstructing the model we have done some further refinement.

Reviewer #2 (Remarks to the Author):

I'm pleased to say I have come to appreciate this manuscript a lot more than I did originally, provided certain constraints are made very obvious to readers.

My original concern was that the functional validation is weak by cancer biology or immunology standards for a journal of this calibre, with no in vivo verification of the molecule/function relationships described, just broadly aligned with previously published in vivo observations. But I take the authors' point in rebuttal that the validation is intended to validate the model's performance rather than define new mechanistic relationships. The value of the work lies in the model, which is interesting, timely and novel and represents significant progress. However, so much of the paper is devoted to its validation, that it is necessary to make this distinction much clearer, and to readers from a broad range of specialities. I happily admit I misjudged its intended significance originally, but I am confident I would not be alone in that, so it is something to consider.

We thank the reviewer for their comment – to ensure that the intent is clearer we have made edits throughout the text, including in the title of the manuscript.

It is important to carefully avoid mechanistic overstatement, because identifying novel regulators of cellular attachment etc in novel cell types and disease states has not been achieved to a sufficient standard in my opinion (and the authors state in rebuttal that describing mechanism this is not their intent). For example, I would like to see the title of the paper mention the predictive model.

We have altered the title to better reflect that the key focus of the manuscript is the model and its validation stems from that (lines 1-3).

Currently it focuses the reader towards molecular function in disease states. Cell lines are not diseases, and the interesting t-SNE work on human cancers is supportive but entirely correlative. The model is the novel aspect of this paper, far more than the effects of ion channel networks on cancer or in cultured FRCs, which is weakly proven. But I appreciate the point that the data should be seen here solely as a system of in vitro validation for the model's predictions. Provided it is clear that these molecular functions are yet to be validated in vivo, I agree that the data fulfils the intended purpose. The writing now needs to better reflect this intention.

We have altered the text to better reflect that the work does not involve in vivo validation as suggested by the reviewer in the following paragraphs, and have increased the emphasis on the model.

The validation is also somewhat weakened because effects of single proteins are mostly viewed in isolation rather than net effects of multiple proteins, which is the model's greatest advantage over literature searches and other traditional single molecule discovery approaches (so its performance in this respect is of great interest). But the work describing successful co-inhibition of SLC4 and SLC9 is an important validation and very interesting. I do still think it should be validated in vivo at least once before we can be confident in the model's performance and future relevance (particularly in light of the title), but I leave that to the discretion of the editorial team.

Whilst we understand the criticism of the author- in the FRCs we made predictions on the effect of individual proteins- we disagree that it necessarily follows that our observations are limited to single proteins. When the model is modified all proteins connected to the modified node may respond, and the effect of the protein may result directly from inhibition of its own specific activity, or through its interactions with downstream effectors. The role of these downstream effectors is further modulated by cell line specific expression, and the response can therefore be expected to vary across cell types and conditions. It follows that in order for phenotypes to be reproduced accurately, the model must correctly incorporate the responses of many proteins to the loss of a single gene. **We have altered the main text to better reflect this this** (lines 467-470)

Minor points:

Line 677: "LPS exposure represents changes associated with the LN immune response, and is an entirely different physiological stimulus to TFs. We find that our model accurately predicted the cellular responses displayed in the literature" – table 3 reports no prediction

from literature for the effect of LPS on FRCs.

Table 3 contains literature predictions under the heading “LPS exposure”, in particular, the table states that cell viability should increase (Fletcher 2015), cell membrane dynamics/movement should increase (Malhotra 2012), Contractility should decrease (Fletcher 2015, Astarita 2015), and Cell Size should increase (Acton 2015).

Line 705: “Whilst we do not claim to completely encapsulate all cellular behaviour, or the only mechanism for its change, we have shown that focussing on ion channels is enough to partly explain cellular morphological change, and provide useful predictions for modulation of cellular behaviour.” – mention of in vitro limitations to the work would improve the interpretation here.

We have altered the sentence to explicitly note the in vitro validation.

Line 668: “Stromal cells are highly integral to this process, ... a mechanism has not previously been elucidated. We propose that the reorganisation of ion channel expression is one of the key mechanisms by which cellular transformation of TDLN FRCs occurs. We have shown that targeted knockdown of certain channels can modulate behaviour in specific ways within FRCs.” – clear mention of in vitro limitations to the work would improve the interpretation here.

We have altered the sentence to better reflect the vitro limitations of the work.

Line 648: “and show that dysregulation of the membrane transport network can explain and impact the behaviour of cells in disease such as cancer.” – “may explain and impact” would be more accurate here.

We have altered the sentence as requested.

Reviewer #3 (Remarks to the Author):

The paper is now suitable for publication.

REVIEWERS' COMMENTS:

Reviewer #4 (Remarks to the Author):

Authors have successfully addressed all the comments raised.

Minor comment: Please fix the link to Bio Model Analyzer. Instead of www it should be https:
<https://biomodelanalyzer.org/>

REVIEWERS' COMMENTS:

Reviewer #4 (Remarks to the Author):

Authors have successfully addressed all the comments raised.

Minor comment: Please fix the link to Bio Model Analyzer. Instead of www it should be https:

<https://biomodelanalyzer.org/>

We apologise for the mistake – we have corrected all references to the BioModelAnalyzer.